# Adaptive Privacy Composition for Accuracy-first Mechanisms

**Ryan Rogers**
LinkedIn

**Gennady Samorodnitsky**
Cornell University

**Zhiwei Steven Wu**
Carnegie Mellon University

**Aaditya Ramdas**
Carnegie Mellon University

## Abstract

In many practical applications of differential privacy, practitioners seek to provide the best privacy guarantees subject to a target level of accuracy. A recent line of work by Ligett et al. [2017], Whitehouse et al. [2022] has developed such accuracy-first mechanisms by leveraging the idea of *noise reduction* that adds correlated noise to the sufficient statistic in a private computation and produces a sequence of increasingly accurate answers. A major advantage of noise reduction mechanisms is that the analysts only pay the privacy cost of the least noisy or most accurate answer released. Despite this appealing property in isolation, there has not been a systematic study on how to use them in conjunction with other differentially private mechanisms. A fundamental challenge is that the privacy guarantee for noise reduction mechanisms is (necessarily) formulated as *ex-post privacy* that bounds the privacy loss as a function of the released outcome. Furthermore, there has yet to be any study on how ex-post private mechanisms compose, which allows us to track the accumulated privacy over several mechanisms. We develop privacy filters [Rogers et al., 2016, Feldman and Zrnic, 2021, Whitehouse et al., 2023] that allow an analyst to adaptively switch between differentially private and ex-post private mechanisms subject to an overall differential privacy guarantee.

## 1 Introduction

Although differential privacy has been recognized by the research community as the de-facto standard to ensure the privacy of a sensitive dataset while still allowing useful insights, it has yet to become widely applied in practice despite its promise to ensure formal privacy guarantees. There are notable applications of differential privacy, including the U.S. Census [Abowd et al., 2022], yet few would argue that differential privacy has become quite standard in practice.

One common objection to differential privacy is that it injects noise and can cause spurious results for data analyses. A recent line of work in differential privacy has focused on developing *accuracy-first* mechanisms that aim to ensure a target accuracy guarantee while achieving the best privacy guarantee [Ligett et al., 2017, Whitehouse et al., 2022]. In particular, these *accuracy-first* mechanisms do not ensure a predetermined level of privacy, but instead provide *ex-post privacy*, which allows the resulting privacy loss to depend on the outcome of the mechanism. This is in contrast to the prevalent paradigm of differential privacy that fixes the scale of privacy noise in advance and hopes the result is accurate. With accuracy-first mechanisms, practitioners instead specify the levels of accuracy that would ensure useful data analyses and then aim to achieve such utility with the strongest privacy guarantee.

However, one of the limitations of this line of work is that it is not clear how ex-post privacy mechanisms *compose*, so if we combine multiple ex-post privacy mechanisms, what is the overall

37th Conference on Neural Information Processing Systems (NeurIPS 2023).

privacy guarantee? Composition is one of the key properties of differential privacy (when used in a *privacy-first* manner), so it is important to develop a composition theory for ex-post privacy mechanisms. Moreover, how do we analyze the privacy guarantee when we compose ex-post privacy mechanisms with differentially private mechanisms?

Our work seeks to answer these questions by connecting with another line work in differential privacy on *fully adaptive privacy composition*. Traditional differential privacy composition results would require the analyst to fix privacy loss parameters, which is then inversely proportional to the scale of noise, for each analysis in advance, prior to any interaction. Knowing that there will be noise, the data scientist may want to select different levels of noise for different analyses, subject to some overall privacy budget. Privacy filters and odometers, introduced in Rogers et al. [2016], provide a way to bound the overall privacy guarantee despite adaptively selected privacy loss parameters. There have since been other works that have improved on the privacy loss bounds in this adaptive setting, to the point of matching (including constants) what one achieves in a nonadaptive setting [Feldman and Zrnic, 2021, Whitehouse et al., 2023].

A natural next step would then be to allow an analyst some overall privacy loss budget to interact with the dataset and the analyst can then determine the accuracy metric they want to set with each new query. As a motivating example, consider some accuracy metric of $\alpha\%$ relative error to different counts with some overall privacy loss parameters $\epsilon, \delta$, so that the entire interaction will be $(\varepsilon, \delta)$-differentially private. The first true count might be very large, so the amount of noise that needs to be added to ensure the target $\alpha\%$ relative error can be huge, and hence very little of the privacy budget should be used for that query, allowing for potentially more results to be returned than an approach that sets an a priori noise level.

A baseline approach to add relative noise would be to add a large amount of noise and then check if the noisy count is within some tolerance based on the scale of noise added, then if the noisy count is deemed good enough we stop, otherwise we scale the privacy loss up by some factor and repeat. We refer to this approach as the *doubling approach* (see Section 7.1 for more details), which was also used in Ligett et al. [2017]. The primary issue with this approach is that the accumulated privacy loss needs to combine the privacy loss each time we add noise, even though we are only interested in the outcome when we stopped. Noise reduction mechanisms from Ligett et al. [2017], Whitehouse et al. [2022] show how it is possible to only pay for the privacy of the last noise addition. However, it is not clear how the privacy loss will accumulate over several noise reduction mechanisms, since each one ensures ex-post privacy, not differential privacy.

We make the following contributions in this work:

- We present a general (basic) composition result for ex-post privacy mechanisms that can be used to create a privacy filter when an analyst can select arbitrary ex-post privacy mechanisms and (concentrated) differentially private mechanisms.
- We develop a unified privacy filter that combines noise reduction mechanisms — specifically the Brownian Mechanism [Whitehouse et al., 2022] — with traditional (concentrated) differentially private mechanisms.
- We apply our results to the task of releasing counts from a dataset subject to a relative error bound comparing the unified privacy filter and the baseline doubling approach, which uses the privacy filters from Whitehouse et al. [2023].

Our main technical contribution is in the unified privacy filter for noise reduction mechanisms and differentially private mechanisms. Prior work [Whitehouse et al., 2022] showed that the privacy loss of the Brownian noise reduction can be written in terms of a scaled Brownian motion at the time where the noise reduction was stopped. We present a new analysis for the ex-post privacy guarantee of the Brownian mechanism that considers a reverse time martingale, based on a scaled standard Brownian motion. Composition bounds for differential privacy consider a forward time martingale and apply a concentration bound, such as Azuma's inequality [Dwork et al., 2010], so we show how we can construct a forward time martingale from the stopped Brownian motions, despite the stopping times being adaptive, not predictable, at each time. See Figure 1 for a sketch of how we combine reverse time martingales into an overall forward time filtration.

There have been other works that have considered adding relative noise subject to differential privacy. In particular, iReduct [Xiao et al., 2011] was developed to release a batch of queries subject to a relative error bound. The primary difference between that work and our setting here is that we do not

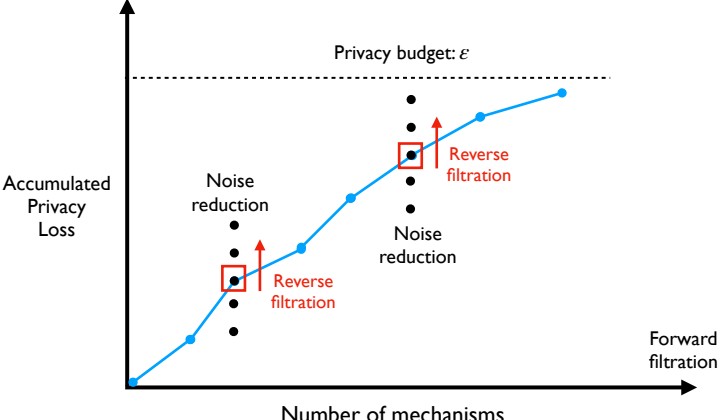

Figure 1: Our privacy filter tracks an accumulated privacy loss over many different mechanisms (forward filtration, X-axis) and stopping when it exceeds a predefined $\epsilon$ (dotted line). Each mechanism satisfies approximate zCDP (blue dot) or is a noise reduction mechanism (black dots). The latter itself involves several rounds of interaction in a reverse filtration (red arrow) until a stopping criterion based on utility is met (red box). Later queries/mechanisms can depend on past ones.

want to fix the number of queries in advance and we want to allow the queries to be selected in an adaptive way. Xiao et al. [2011] consider a batch of $m$ queries and initially adds a lot of noise to each count and iteratively checks whether the noisy counts are good enough. The counts that should have smaller noise are then identified and a Laplace based noise reduction algorithm is used to decrease the noise on the identified set. They continue in this way until either all counts are good enough or a target privacy loss is exhausted. There are two scenarios that might arise: (1) all counts satisfy the relative error condition and we should have tried more counts because some privacy loss budget remains, (2) the procedure stopped before some results had a target relative error, so we should have selected fewer queries. In either case, selecting the number of queries becomes a parameter that the data analyst would need to select in advance, which would be difficult to do a priori. In our setting, no such parameter arises. Furthermore, they add up the privacy loss parameters for each count to see if it is below a target privacy loss bound at each step (based on the $\ell_1$-general sensitivity), however we show that adding up the privacy loss parameters can be significantly improved on. Other works have modified the definition of differential privacy to accommodate relative error, e.g. Wang et al. [2022].

## 2 Preliminaries

We start with some basic definitions from differential privacy, beginning with the standard definition from Dwork et al. [2006b,a]. We first need to define what we mean by *neighboring datasets*, which can mean adding or removing one record to a dataset or changing an entry in a dataset. We will leave the neighboring relation arbitrary, and write $x, x' \in \mathcal{X}$ to be neighbors as $x \sim x'$.

**Definition 2.1.** *An algorithm $A : \mathcal{X} \to \mathcal{Y}$ is $(\epsilon, \delta)$-differentially private if, for any measurable set $E \subset \mathcal{Y}$ and any neighboring inputs $x \sim x'$, $\Pr[A(x) \in E] \leq e^\epsilon \Pr[A(x') \in E] + \delta$. If $\delta = 0$, we say $A$ is $\varepsilon$-DP or simply* pure DP.

We define a statistic's *sensitivity* as the following: $\Delta_p(f) := \max_{x,x':x \sim x'} \{||f(x) - f(x')||_p\}$.

A central actor in our analysis will be the *privacy loss* of an algorithm, which will be a random variable that depends on the outcomes of the algorithm under a particular dataset.

**Definition 2.2** (Privacy Loss). *Let $A : \mathcal{X} \to \mathcal{Y}$ be an algorithm and fix neighbors $x \sim x'$ in $\mathcal{X}$. Let $p^x$ and $p^{x'}$ be the respective densities of $A(x)$ and $A(x')$ on the space $\mathcal{Y}$ with respect to some reference measure. Then, the privacy loss between $A(x)$ and $A(x')$ evaluated at point $y \in \mathcal{Y}$ is:*

$$\mathcal{L}_A(y; x, x') := \log\left(\frac{p^x(y)}{p^{x'}(y)}\right).$$

*Further, we refer to the privacy loss random variable to be $\mathcal{L}_A(x, x') := \mathcal{L}_A(A(x); x, x')$. When the algorithm A and the neighboring datasets are clear from context, we drop them from the privacy loss, i.e. $\mathcal{L}(y) = \mathcal{L}_A(y; x, x')$ and $\mathcal{L} = \mathcal{L}_A(x, x')$.*

Many existing privacy composition analyses leverage a variant of DP called (approximate) *zero-concentrated DP* (zCDP), introduced by Bun and Steinke [2016]. Recall that the Rényi Divergence of order $\lambda \geq 1$ between two distributions $P$ and $Q$ on the same domain is written as the following where $p(\cdot)$ and $q(\cdot)$ are the respective probability mass/density functions,

$$D_\lambda(P\|Q) := \frac{1}{\lambda - 1} \log \left( \mathbb{E}_{y \sim P} \left[ \left( \frac{p(y)}{q(y)} \right)^{\lambda - 1} \right] \right).$$

Since we study fully adaptive privacy composition (where privacy parameters can be chosen adaptively), we will use the following *conditional* extension of approximate zCDP, in which the zCDP parameters of a mechanism $A$ can depend on prior outcomes.

**Definition 2.3** (Approximate zCDP [Whitehouse et al., 2023]). *Suppose $A : \mathcal{X} \times \mathcal{Z} \to \mathcal{Y}$ with outputs in a measurable space $(\mathcal{Y}, \mathcal{G})$. Suppose $\delta, \rho : \mathcal{Y} \to \mathbb{R}_{\geq 0}$. We say the algorithm $A$ satisfies conditional $\delta(z)$-approximate $\rho(z)$-zCDP if, for all $z \in \mathcal{Z}$ and any neighboring datasets $x, x'$, there exist probability transition kernels $P', P'', Q', Q'' : \mathcal{Z} \times \mathcal{G} \to [0, 1]$ such that the conditional outputs are distributed according to the following mixture distributions:*

$$A(x; z) \sim (1 - \delta(z))P'(\cdot \mid z) + \delta(z)P''(\cdot \mid z)$$
$$A(x'; z) \sim (1 - \delta(z))Q'(\cdot \mid z) + \delta(z)Q''(\cdot \mid z),$$

*where for all $\lambda \geq 1$, $D_\lambda(P'(\cdot \mid z)\|Q'(\cdot \mid z)) \leq \rho(z)\lambda$ and $D_\lambda(Q'(\cdot \mid z)\|P'(\cdot \mid z)) \leq \rho(z)\lambda$, $\forall z \in \mathcal{Z}$.*

The following results establish the relationship between zCDP and DP and the composition of zCDP.

**Lemma 2.1** (Bun and Steinke [2016]). *If $M : \mathcal{X} \to \mathcal{Y}$ is $(\varepsilon, \delta)$-DP, then $M$ is $\delta$-approximate $\varepsilon^2/2$-zCDP. If $M$ is $\delta$-approximate $\rho$-zCDP, then $M$ is $(\rho + 2\sqrt{\rho \log(1/\delta'')}, \delta + \delta'')$-DP for $\delta'' > 0$.*

**Lemma 2.2** (Bun and Steinke [2016]). *If $M_1 : \mathcal{X} \to \mathcal{Y}$ is $\delta_1$-approximate $\rho_1$-zCDP and $M_2 : \mathcal{X} \times \mathcal{Y} \to \mathcal{Y}'$ is $\delta_2$-approximate $\rho_2$-zCDP in its first coordinate, then $M : \mathcal{X} \to \mathcal{Y}'$ where $M(x) = (M_1(x), M_2(x, M_1(x)))$ is $(\delta_1 + \delta_2)$-approximate $(\rho_1 + \rho_2)$-zCDP.*

## 3 Privacy Filters

In order for us to reason about the overall privacy loss of an interaction with a sensitive dataset, we will use the framework of *privacy filters*, introduced in Rogers et al. [2016]. Privacy filters allow an analyst to adaptively select privacy parameters as a function of previous outcomes until some stopping time that determines whether a target privacy budget has been exhausted. To denote that privacy parameters may depend on previous outcomes, we will write $\rho_n(x)$ to mean the privacy parameter selected at round $n$ that could depend on the previous outcomes $A_{1:n-1}(x)$, and similarly for $\delta_n(x)$. We now state the definition of privacy filters in the context of approximate zCDP mechanisms.

**Definition 3.1** (Privacy Filter). *Let $(A_n : \mathcal{X} \to \mathcal{Y})_{n \geq 1}$ be an adaptive sequence of algorithms such that, for all $n \geq 1$, $A_n(\cdot; y_{1:n-1})$ is $\delta_n(y_{1:n-1})$-approximate $\rho_n(y_{1:n-1})$-zCDP for all $y_{1:n-1} \in \mathcal{Y}^{n-1}$. Let $\epsilon > 0$ and $\delta \geq 0$ be fixed privacy parameters. Then, a function $N : \mathcal{Y}^\infty \to \mathbb{N}$ is an $(\epsilon, \delta)$-privacy filter if*

1. *for all $(y_1, y_2, \cdots) \in \mathcal{Y}^\infty$, $N(y_1, y_2, \cdots)$ is a stopping time with respect to the natural filtration generated by $(A_n(x))_{n \geq 1}$, and*

2. *$A_{1:N(\cdot)}(\cdot)$ is $(\epsilon, \delta)$-differentially private where $N(x) = N(A_1(x), A_2(x), \cdots)$.*

Whitehouse et al. [2023] showed that we can use composition bounds from traditional differential privacy (which required setting privacy parameters for the interaction prior to any interaction) in the more adaptive setting, where privacy parameters can be set before each query.

**Theorem 1.** *Suppose $(A_n : \mathcal{X} \to \mathcal{Y})_{n \geq 1}$ is a sequence of algorithms such that, for any $n \geq 1$, $A_n(\cdot; y_{1:n-1})$ is $\delta_n(y_{1:n-1})$-approximate $\rho_n(y_{1:n-1})$-zCDP for all $y_{1:n-1}$. Let $\epsilon > 0$ and $\delta \geq 0$,*

$\delta'' > 0$ *be fixed privacy parameters. Consider the function* $N : \mathbb{R}_{\geq 0}^{\infty} \to \mathbb{N}$ *given by the following where* $\delta < \sum_{m \leq n+1} \delta_m(y_{1:m-1})$ *for all* $y_1, y_2, \cdots$ *and*

$$N(y_1, y_2, \cdots) := \inf \left\{ n : \epsilon < 2\sqrt{\log\left(\frac{1}{\delta''}\right) \sum_{m \leq n+1} \rho_m(y_{1:m-1})} + \sum_{m \leq n+1} \rho_m(y_{1:m-1}) \right\}.$$

*Then, the algorithm* $A_{1:N(\cdot)}(\cdot)$ *is* $(\epsilon, \delta + \delta'')$-*DP, where* $N(x) := N((A_n(x))_{n \geq 1})$. *In other words,* $N$ *is an* $(\epsilon, \delta)$-*privacy filter.*

## 4 Ex-post Private Mechanisms

Although privacy filters allow a lot of flexibility to a data analyst in how they interact with a sensitive dataset while still guaranteeing a fixed privacy budget, there are some algorithms that ensure a bound on privacy that is adapted to the dataset. Ex-post private mechanisms define privacy loss as a probabilistic bound which can depend on the algorithm's outcomes, so some outcomes might contain more information about an individual than others [Ligett et al., 2017, Whitehouse et al., 2022]. Note that ex-post private mechanisms do not have any fixed a priori bound on the privacy loss, so by default they cannot be composed in a similar way to differentially private mechanisms.

**Definition 4.1.** *Let* $A : \mathcal{X} \times \mathcal{Y} \to \mathcal{Z}$ *be an algorithm and* $\mathcal{E} : \mathcal{Z} \times \mathcal{Y} \to \mathbb{R}_{\geq 0}$ *a function. We say* $A(\cdot \, ; y)$ *is* $(\mathcal{E}(\cdot \, ; y), \delta(y))$-*ex-post private for all* $y \in \mathcal{Y}$ *if, for any neighboring inputs* $x \sim x'$, *we have* $\Pr[\mathcal{L}(A(x; y)) > \mathcal{E}(A(x; y); y)] \leq \delta(y)$ *for all* $y \in \mathcal{Y}$.

We next define a single noise reduction mechanism, which will interactively apply sub-mechanisms and stop at some time $k$, which can be random. Each iterate will use a privacy parameter from an increasing sequence of privacy parameters $(\varepsilon^{(n)} : n \geq 1)$ set in advance and the overall privacy will only depend on the last privacy parameter $\epsilon^{(k)}$, despite releasing noisy values with parameters $\varepsilon^{(i)}$ for $i \leq k$. Noise reduction algorithms will allow us to form ex-post private mechanisms because the privacy loss will only depend on the final outcome. We will write $M : \mathcal{X} \to \mathcal{Y}^*$ to be any algorithm mapping databases to sequences of outputs in $\mathcal{Y}$, with intermediate mechanisms written as $M^{(k)} : \mathcal{X} \to \mathcal{Y}$ for the $k$-th element of the sequence and $M^{(1:k)} : \mathcal{X} \to \mathcal{Y}^k$ for the first $k$ elements. Let $\mu$ be a probability measure on $\mathcal{Y}$, for each $k \geq 1$ let $\mu_k$ be a probability measure on $\mathcal{Y}^k$, and let $\mu^*$ be a probability measure on $\mathcal{Y}^*$. We assume that the law of $M^{(k)}(x)$ on $\mathcal{Y}$ is equivalent to $\mu$, the law of $M(x)$ on $\mathcal{Y}^*$ is equivalent to $\mu^*$, and the law of $M^{(1:k)}(x)$ on $\mathcal{Y}^k$ is equivalent to $\mu_k$ for every $k$ and every $x$. Furthermore, we will write $\mathcal{L}$ to be the privacy loss of $M$, $\mathcal{L}^{(k)}$ be the privacy loss of $M^{(k)}$, and $\mathcal{L}^{(1:k)}$ to be the privacy loss of the sequence of mechanisms $M^{(1:k)}$. We then define noise reduction mechanisms more formally.

**Definition 4.2** (Noise Reduction Mechanism). *Let* $M : \mathcal{X} \to \mathcal{Y}^{\infty}$ *be a mechanism mapping sequences of outcomes and* $x, x'$ *be any neighboring datasets. We say* $M$ *is a* noise reduction mechanism *if for any* $k \geq 1$, $\mathcal{L}^{(1:k)} = \mathcal{L}^{(k)}$.

We will assume there is a fixed grid of time values $t^{(1)} > t^{(2)} > \cdots > t^{(k)} > 0$. We will typically think of the time values as being inversely proportional to the noise we add, i.e. $t^{(i)} = \left(1/\varepsilon^{(i)}\right)^2$ where $\varepsilon^{(1)} < \varepsilon^{(2)} < \cdots$. An analyst will not have a particular stopping time set in advance and will instead want to stop interacting with the dataset as a function of the noisy answers that have been released. It might also be the case that the analyst wants to stop based on the outcome and the privatized dataset, but in this work we consider stopping times that can only depend on the noisy outcomes or possibly some public information, not the underlying dataset.

**Definition 4.3** (Stopping Function). *Let* $A : \mathcal{X} \to \mathcal{Y}^{\infty}$ *be a noise reduction mechanism. For* $x \in \mathcal{X}$, *let* $(\mathcal{F}^{(k)}(x))_{k \in \mathbb{N}}$ *be the filtration given by* $\mathcal{F}^{(k)}(x) := \sigma(A^{(i)}(x) : i \leq k)$. *A function* $T : \mathcal{Y}^{\infty} \to \mathbb{N}$ *is called a stopping function if for any* $x \in \mathcal{X}$, $T(x) := T(A(x))$ *is a stopping time with respect to* $(\mathcal{F}^{(k)}(x))_{k \geq 1}$. *Note that this property does not depend on the choice of measures* $\mu$, $\mu^*$ *and* $\mu_k$.

We now recall the noise reduction mechanism with Brownian noise [Whitehouse et al., 2022].

**Definition 4.4** (Brownian Noise Reduction). *Let* $f : \mathcal{X} \to \mathbb{R}^d$ *be a function and* $(t^{(k)})_{k \geq 1}$ *be a sequence of time values. Let* $(B^{(t)})_{t \geq 0}$ *be a standard* $d$-*dimensional Brownian motion and* $T :$

$(\mathbb{R}^d)^\infty \to \mathbb{N}$ *be a stopping function. The Brownian mechanism associated with $f$, time values $(t^{(k)})_{k \geq 1}$, and stopping function $T$ is the algorithm* $\texttt{BM} : \mathcal{X} \to ((0, t^{(1)}] \times \mathbb{R}^d)^*$ *given by*

$$\texttt{BM}(x) := \left( t^{(k)}, f(x) + B^{(t^{(k)})} \right)_{k \leq T(x)}.$$

We then have the following result.

**Lemma 4.1** (Whitehouse et al. [2022]). *The Brownian Noise Reduction mechanism* $\texttt{BM}$ *is a noise reduction algorithm for a constant stopping function $T(x) = k$. Furthermore, we have for any stopping function $T : \mathcal{Y}^\infty \to \mathbb{N}$, the noise reduction property still holds, i.e.* $\mathcal{L}^{(1:T(x))} = \mathcal{L}^{(T(x))}$.

Another noise reduction mechanism uses Laplace noise from Koufogiannis et al. [2017], Ligett et al. [2017], which we consider in Appendix C.

# 5 General Composition of Ex-post Private Mechanisms

We consider combining zCDP mechanisms with mechanisms that satisfy ex-post privacy. We consider a sequence of mechanisms $(A_n : \mathcal{X} \to \mathcal{Y})_{n \geq 1}$ where each mechanism may depend on the previous outcomes. At each round, an analyst will use either an ex-post private mechanism or an approximate zCDP mechanism, in either case the privacy parameters may depend on the previous results as well. We will write $\delta_n(x) := \delta_n(A_1(x), \cdots, A_{n-1}(x))(x))$, $\rho_n(x) := \rho(A_1(x), \cdots, A_{n-1}(x))(x))$, and $\mathcal{E}_n(A_n(x); x) := \mathcal{E}_n(A_n(x); A_1(x), \cdots, A_{n-1}(x))$.

**Definition 5.1** (Approximate zCDP and Ex-post Private Sequence). *Consider a sequence of mechanisms $(A_n)_{n \geq 1}$, where $A_n : \mathcal{X} \to \mathcal{Y}$. The sequence $(A_n)_{n \geq 1}$ is called a sequence of* approximate zCDP and ex-post private mechanisms *if for each round $n$, the analyst will select $A_n(\cdot)$ to be $\delta_n(\cdot)$-approximate $\rho_n(\cdot)$-zCDP given previous outcomes $A_i(\cdot)$ for $i < n$, or the analyst will select $A_n(\cdot)$ to be $(\mathcal{E}_n(A_n(\cdot); \cdot), \delta'_n(\cdot))$-ex-post private conditioned on $A_i(\cdot)$ for $i < n$. In rounds where zCDP is selected, we will simply write $\mathcal{E}_i(A_i(\cdot); \cdot) \equiv 0$, while in rounds where an ex-post private mechanism is selected, we will set $\rho_i(\cdot) = 0$.*

We now state a composition result that allows an analyst to combine ex-post private and zCDP mechanisms adaptively, while still ensuring a target level of privacy. Because we have two different interactive systems that are differentially private, one that uses only zCDP mechanisms and the other that only uses ex-post private mechanisms, we can then use concurrent composition, see Appendix A for more details.

**Theorem 2.** *Let $\varepsilon, \varepsilon', \delta, \delta', \delta'' > 0$. Let $(A_n)_{n \geq 1}$ be a sequence of approximate zCDP and ex-post private mechanisms. We require that the ex-post private mechanisms that are selected at each round $n$ to have ex-post privacy functions $\mathcal{E}_n$ that do not exceed the remaining budget from $\epsilon'$. Consider the following function $N : \mathcal{Y}^\infty \to \mathbb{N}$ as the following for any sequence of outcomes $(y_n)_{n \geq 1}$*

$$N((y_n)_{n \geq 1}, (y_n) = \inf \left\{ N_{\text{zCDP}}((y_{1:n-1})_{n \geq 1}), N_{\text{post}}((y_n)_{n \geq 1}) \right\},$$

*where $N_{\text{zCDP}}((y_n)_{n \geq 1})$ is the stopping rule given in Theorem 1 with privacy parameters $\varepsilon, \delta, \delta''$ and $N_{\text{post}}((y_n)_{n \geq 1})$ is the stopping rule where the sum of realized $\mathcal{E}_n(y_n; y_1, \cdots, y_{n-1})$ values cannot go above $\varepsilon'$ nor can the sum of $\delta'_n(y_1, \cdots, y_{n-1})$ go above $\delta'$ (see Lemma B.2 with privacy parameters $\varepsilon', \delta'$). Then, the algorithm $A_{1:N(\cdot)}(\cdot)$ is $(\varepsilon + \varepsilon', \delta + \delta' + \delta'')$-DP, where $N(x) = N((A_n(x))_{n \geq 1})$.*

Although we are able to leverage *advanced composition* bounds from traditional differential privacy for the mechanisms that are approximate zCDP, we are simply adding up the ex-post privacy guarantees, which seems wasteful. Next, we consider how we can improve on this composition bound for certain ex-post private mechanisms.

# 6 Brownian Noise Reduction Mechanisms

We now consider composing specific types of ex-post private mechanisms, specifically the Brownian Noise Reduction mechanism. From Theorem 3.4 in Whitehouse et al. [2022], we can decompose the privacy loss as an uncentered Brownian motion, even when the stopping time is adaptively selected.

**Theorem 3** (Whitehouse et al. [2022]). *Let BM be the Brownian noise reduction mechanism associated with time values $(t^{(k)})_{k \geq 1}$ and a function $f$. All reference measures generated by the mechanism are those generated by the Brownian motion without shift (starting at $f(x) = 0$). For neighbors $x \sim x'$ and stopping function $T$, the privacy loss between $BM^{(1:T(x))}(x)$ and $BM^{(1:T(x'))}(x')$ is given by*

$$\mathcal{L}_{BM}^{(1:T(x))}(x, x') = \frac{||f(x') - f(x)||_2^2}{2t^{(T(x))}} + \frac{||f(x') - f(x)||_2}{t^{(T(x))}} \left\langle \frac{f(x') - f(x)}{||f(x') - f(x)||_2}, B^{(t^{(T(x))})} \right\rangle.$$

This decomposition of the privacy loss will be very useful in analyzing the overall privacy loss of a combination of Brownian noise reduction mechanisms. In Appendix C we provide a more general analysis than in Whitehouse et al. [2022] to show that both the Laplace noise reduction from Ligett et al. [2017] and the Brownian noise reduction Whitehouse et al. [2022] are indeed noise reduction mechanisms and give the form of the privacy loss at an adaptive stopping time.

## 6.1 Backward Brownian Motion Martingale

We now present a key result for our analysis of composing Brownian noise reduction mechanisms. Although in Whitehouse et al. [2022], the ex-post privacy proof of the Brownian mechanism applied Ville's inequality [for proof, cf. Howard et al., 2020, Lemma 1] to the (unscaled) standard Brownian motion $(B^{(t)})_{t>0}$, it turns out that the scaled standard Brownian motion $(B^{(t)}/t)$ forms a backward martingale [cf. Revuz and Yor, 2013, Exercise 2.16] and this fact is crucial to our analysis.

**Lemma 6.1** (Backward martingale). *Let $(B^{(t)})$ be a standard Brownian motion. Define the reverse filtration $\mathcal{G}^{(t)} = \sigma(B^{(u)}; u \geq t)$, meaning that $\mathcal{G}^{(s)} \supset \mathcal{G}^{(t)}$ if $s < t$. For every real $\lambda$, the process*

$$M^{(t)} := \exp(\lambda B^{(t)}/t - \lambda^2/(2t)); \quad t > 0$$

*is a nonnegative (reverse) martingale with respect to the filtration $\mathcal{G} = (\mathcal{G}^{(t)})$. Further, at any $t > 0$, $\mathbb{E}[M^{(t)}] = 1$, $M^{(\infty)} = 1$ almost surely, and $\mathbb{E}[M^{(\tau)}] \leq 1$ for any stopping time $\tau$ with respect to $\mathcal{G}$ (equality holds with some restrictions). In short, $M^{(\tau)}$ is an "e-value" for any stopping time $\tau$ — an e-value is a nonnegative random variable with expectation at most one.*

Let $B_1 = (B_1^{(t)})_{t \geq 0}, B_2 = (B_2^{(t)})_{t \geq 0}, \ldots$ be independent, standard Brownian motions, with corresponding backward martingales $M_1 = (M_1^{(t)})_{t \geq 0}, M_2 = (M_2^{(t)})_{t \geq 0}, \ldots$ and (internal to each Brownian motion) filtrations $\mathcal{G}_1 = (\mathcal{G}_1^{(t)})_{t \geq 0}, \mathcal{G}_2 = (\mathcal{G}_2^{(t)})_{t \geq 0}, \ldots$ as defined in the previous lemma. Select time values $(t_1^{(k)})_{k \geq 1}$. Let a Brownian noise reduction mechanism $BM_1$ be run using $B_1$ and stopped at $\tau_1$. Then $E[M_1^{(\tau_1)}] \leq 1$ as per the previous lemma. Based on outputs from the $BM_1$, we choose time values $(t_2^{(k)})_{k \geq 1} \in \sigma(B_1^{(t)}; t \geq \tau_1) := \mathcal{F}_1$. Now run the second Brownian noise reduction $BM_2$ using $B_2$, stopping at time $\tau_2$. Since $B_1, B_2$ are independent, we still have that $\mathbb{E}[M_2^{(\tau_2)}|\mathcal{F}_1] \leq 1$. Let $\mathcal{F}_2 := \sigma((B_1^{(t)})_{t \geq \tau_1}, (B_2^{(t)})_{t \geq \tau_2})$ be the updated filtration, based on which we choose time values $(t_3^{(k)})_{k \geq 1}$. Because $B_3$ is independent of the earlier two, at the next step, we still have $\mathbb{E}[M_3^{(\tau_3)}|\mathcal{F}_2] \leq 1$. Proceeding in this fashion, it is clear that the product of the stopped e-values $E_m$, where

$$E_m := \prod_{s=1}^{m} M_s^{(\tau_s)} = \exp\left(\lambda \sum_{s=1}^{m} \frac{B_s^{(\tau_s)}}{\tau_s} - \frac{\lambda^2}{2} \sum_{s=1}^{m} \frac{1}{\tau_s}\right), \tag{1}$$

is itself a (forward) nonnegative supermartingale with respect to the filtration $\mathcal{F} = (\mathcal{F}_n)_{n \geq 1}$, with initial value $E_0 := 1$. Applying the Gaussian mixture method [cf. Howard et al., 2021, Proposition 5], we get that for any $\gamma, \delta > 0$ and with $\psi(t; \gamma, \delta) := \sqrt{(t + \gamma) \log\left(\frac{t+\gamma}{\delta^2 \gamma}\right)}$,

$$\Pr\left[\sup_{m \geq 1}\left\{\left|\sum_{s=1}^{m} \frac{B_s^{(\tau_s)}}{\tau_s}\right| \geq \sum_{s \in [m]} \frac{1}{2\tau_s} + \psi\left(\sum_{s \in [m]} 1/\tau_s; \gamma, \delta\right)\right\}\right] \leq \delta.$$

This then provides an alternate way to prove Theorem 3.6 in Whitehouse et al. [2022].

**Theorem 4.** *Let $(T_i)_{i \geq 1}$ be a sequence of stopping functions, as in Definition 4.3, and a sequence of time values $(t_i^{(j)} : j \in [k_i])_{i \geq 1}$. Let $BM_i$ denote a Brownian noise reduction with statistic $f_i$ that can be adaptively selected based on outcomes of previous Brownian noise reductions and $f_i$ has $\ell_2$-sensitivity 1. We then have, for any $\gamma, \delta > 0$,*

$$\sup_{x \sim x'} \Pr \left[ \sum_{i=1}^{\infty} \mathcal{L}_{BM_i}^{(T_i(x))} \geq \psi \left( \sum_{i=1}^{\infty} 1/t_i^{(T_i(x))}; \gamma, \delta \right) \right] \leq \delta.$$

*In other words, $\left( BM_i^{(1:T_i(\cdot))} \right)_{i \geq 1}$ is $\left( \psi \left( \sum_{i=1}^{\infty} 1/t_i^{(T_i(\cdot))}; \gamma, \delta \right), \delta \right)$-ex post private.*

## 6.2 Privacy Filters with Brownian Noise Reduction Mechanisms

Given Lemma 6.1 and the decomposition of the privacy loss for the Brownian mechanism given in Theorem 3, we will be able to get tighter composition bounds of multiple Brownian noise reduction mechanisms rather than resorting to the general ex-post privacy composition (see Lemma B.2 in the appendix). It will be important to only use time values with the Brownian noise reduction mechanisms that cannot exceed the remaining privacy loss budget. We then make the following condition on the time values $(t_n^{(j)})_{j=1}^{k_n}$ that are used for each Brownian noise reduction given prior outcomes $y_1, \cdots, y_{n-1}$ from the earlier Brownian noise reductions with time values $(t_i^{(j)})_{j=1}^{k_i}$ and stopping functions $T_i$ for $i < n$ and overall budget $\rho > 0$

$$\frac{1}{2t_n^{(k_n)}} \leq \rho - \sum_{i<n} \frac{1}{2t_i^{(T_i(y_{1:i}))}}. \tag{2}$$

**Lemma 6.2.** *Let $\rho > 0$ and consider a sequence of $(BM_n)_{n \geq 1}$ each with statistic $f_n : \mathcal{X} \to \mathbb{R}^{d_n}$ with $\ell_2$ sensitivity 1, stopping function $T_n$, and time values $(t_n^{(j)})_{j=1}^{k_n}$ which can be adaptively selected and satisfies (2). Consider the function $N : \mathcal{Y}^{\infty} \to \mathbb{N} \cup \{\infty\}$ where $\mathcal{Y}$ contains all possible outcome streams from $(BM_n)_{n \geq 1}$ as the following for any sequence of outcomes $(y_n)_{n \geq 1}$:*

$$N((y_n)_{n \geq 1}) = \inf \left\{ n : \rho = \sum_{i \in [n]} \frac{1}{2t_i^{(T_i(y_{1:i}))}} \right\}.$$

*Then, for $\delta > 0$, $BM_{1:N(\cdot)}(\cdot)$ is $(\rho + 2\sqrt{\rho \log(1/\delta)}, \delta)$-DP, where $N(x) = N((BM_n(x))_{n \geq 1})$.*

One approach to defining a privacy filter for both approximate zCDP and Brownian noise reduction mechanisms would be to use concurrent composition, as we did in Lemma B.2 in the appendix. However, this would require us to set separate privacy budgets for approximate zCDP mechanisms and Brownian noise reduction, which is an extra (nuissance) parameter to set.

We now show how we can combine Brownian noise reduction and approximate zCDP mechanisms with a single privacy budget. We will need a similar condition on the time values selected at each round for the Brownian noise reduction mechanisms as in (2). Note that at each round either an approximate zCDP or Brownian noise reduction mechanism will be selected. Given prior outcomes $y_1, \cdots, y_{n-1}$ and previously selected zCDP parameters $\rho_1, \rho_2(y_1), \cdots, \rho_n(y_{1:n-1})$ — noting that at round $n$ where BM is selected we have $\rho_n(y_{1:n-1}) = 0$ or if a zCDP mechanism is selected we simply set $k_n = 1$ and $\frac{1}{t_n^{(1)}} = 0$ — we have the following condition on $\rho_n(y_{1:n-1})$ and the time values $(t_n^{(j)})_{j=1}^{k_n}$ if we select a BM at round $n$ and have overall budget $\rho > 0$,

$$0 \leq \frac{1}{2t_n^{(k_n)}} + \rho_n(y_{1:n-1}) \leq \rho - \sum_{i<n} \left( \rho_i(y_{1:i-1}) + \frac{1}{2t_i^{(T_i(y_{1:i}))}} \right). \tag{3}$$

**Theorem 5.** *Let $\rho > 0$ and $\delta \geq 0$. Let $(A_n)_{n \geq 1}$ be a sequence of approximate zCDP and ex-post private mechanisms where each ex-post private mechanism at round $n$ is a Brownian Mechanism with an adaptively chosen stopping function $T_n$, a statistic $f_n$ with $\ell_2$-sensitivity equal to 1, and time values $(t_n^{(j)})_{j=1}^{k_n}$ that satisfy the condition in (3). Consider the function $N : \mathcal{Y}^{\infty} \to \mathbb{N}$ as the*

*following for any sequence of outcomes* $(y_n)_{n \geq 1}$ :

$$N((y_n)_{n \geq 1}) = \inf \left\{ n : \delta < \sum_{i \in [n+1]} \delta_i(y_{1:i-1}) \; or \; \rho = \sum_{i \in [n]} \left( \rho_i(y_{1:i-1}) + \frac{1}{2t_i^{(T_i(y_{1:i}))}} \right) \right\}.$$

*Then for $\delta'' > 0$, the algorithm $A_{1:N(\cdot)}(\cdot)$ is $(\rho + 2\sqrt{2\rho \log(1/\delta'')}, \delta + \delta'')$-DP, where the stopping function is $N(x) = N((A_n(x))_{n \geq 1})$.*

## 7  Application: Bounding Relative Error

Our motivating application will be returning as many counts as possible subject to each count being within $\alpha\%$ relative error, i.e. if $y$ is the true count and $\hat{y}$ is the noisy count, we require $||\hat{y}/y| - 1| < \alpha$. It is common for practitioners to be able to tolerate some relative error to some statistics and would like to not be shown counts that are outside a certain accuracy. Typically, DP requires adding a predetermined standard deviation to counts, but it would be advantageous to be able to add large noise to large counts so that more counts could be returned subject to an overall privacy budget.

### 7.1  Doubling Method

A simple baseline approach would be to use the "doubling method", as presented in Ligett et al. [2017]. This approach uses differentially private mechanisms and checks whether each outcome satisfies some condition, in which case you stop, or the analyst continues with a larger privacy loss parameter. The downside of this approach is that the analyst needs to pay for the accrued privacy loss of all the rejected values. However, handling composition in this case turns out to be straightforward given Theorem 1, due to Whitehouse et al. [2023]. We then compare the 'doubling method" against using Brownian noise reduction and applying Theorem 5.

We now present the doubling method formally. We take privacy loss parameters $\varepsilon^{(1)} < \varepsilon^{(2)} < \cdots$, where $\varepsilon^{(i+1)} = \sqrt{2}\varepsilon^{(i)}$. Similar to the argument in Claim B.1 in Ligett et al. [2017], we use the $\sqrt{2}$ factor because the privacy loss will depend on the sum of square of privacy loss parameters, i.e. $\sum_{i=1}^{m} (\varepsilon^{(i)})^2$ up to some iterate $m$, in Theorem 1 as $\rho_i = (\varepsilon^{(i)})^2/2$ is the zCDP parameter. This means that if $\varepsilon^{\star}$ is the privacy loss parameter that the algorithm would have halted at, then we might overshoot it by $\sqrt{2}\varepsilon^{\star}$. Further, the overall sum of square privacy losses will be no more than $4(\varepsilon^{\star})^2$. Hence, we refer to the doubling method as doubling the square of the privacy loss parameter.

### 7.2  Experiments

We perform experiments to return as many results subject to a relative error tolerance $\alpha$ and a privacy budget $\varepsilon, \delta > 0$. We will generate synthetic data from a Zipf distribution, i.e. from a density $f(k) \propto k^{-a}$ for $a > 0$ and $k \in \mathbb{N}$. We will set a max value of the distribution to be $300$ and $a = 0.75$. We will assume that a user can modify each count by at most 1, so that the $\ell_0$-sensitivity is 300 and $\ell_\infty$-sensitivity is 1. See Figure 3 in Appendix B for the data distribution we used.

In our experiments, we will want to first find the top count and then try to add the least amount of noise to it, while still achieving the target relative error, which we set to be $\alpha = 10\%$. To find the top count, we apply the Exponential Mechanism [McSherry and Talwar, 2007] by adding Gumbel noise with scale $1/\varepsilon_{\text{EM}}$ to each sensitivity 1 count (all 300 elements' counts, even if they are zero in the data) and take the element with the top noisy count. From Cesar and Rogers [2021], we know that the Exponential Mechanism with parameter $\varepsilon_{\text{EM}}$ is $\varepsilon_{\text{EM}}^2/8$-zCDP, which we will use in our composition bounds. For the Exponential Mechanism, we select a fixed parameter $\varepsilon_{\text{EM}} = 0.1$.

After we have selected a top element via the Exponential Mechanism, we then need to add some noise to it in order to return its count. Whether we use the doubling method and apply Theorem 1 for composition or the Brownian noise reduction mechanism and apply Theorem 5 for composition, we need a stopping condition. Note that we cannot use the true count to determine when to stop, but we can use the noisy count and the privacy loss parameter that was used. Hence we use the following condition based on the noisy count $\hat{y}$ and the corresponding privacy loss parameter $\varepsilon^{(i)}$ at iterate $i$:

$$1 - \alpha < \left| (\hat{y} + 1/\varepsilon^{(i)})/(\hat{y} - 1/\varepsilon^{(i)}) \right| \leq 1 + \alpha \quad \text{and} \quad |\hat{y}| > 1/\varepsilon^{(i)}.$$

Note that for Brownian noise reduction mechanism at round $n$, we use time values $t_n^{(i)} = 1/(\varepsilon_n^{(i)})^2$. We also set an overall privacy budget of $(\varepsilon, \delta'') = (10, 10^{-6})$. To determine when to stop, we will simply consider the sum of squared privacy parameters and stop if it is more than roughly 2.705, which corresponds to the overall privacy budget of $\varepsilon = 10$ with $\delta'' = 10^{-6}$. If the noisy value from the largest privacy loss parameter does not satisfy the condition above, we discard the result.

We pick the smallest privacy parameter squared to be $(\varepsilon_n^{(1)})^2 = 0.0001$ for each $n$ in both the noise reduction and the doubling method and the largest value will change as we update the remaining sum of square privacy loss terms that have been used. We then set 1000 equally spaced parameters in noise reduction to select between $0.0001$ and the largest value for the square of the privacy loss parameter. We then vary the sample size of the data in $\{8000, 16000, 32000, 64000, 128000\}$ and see how many results are returned and of those returned, how many satisfy the actual relative error, which we refer to as *precision*. Note that if 0 results are returned, then we consider the precision to be 1. Our results are given in Figure 2 where we give the empirical average and standard deviation over 1000 trials for each sample size.

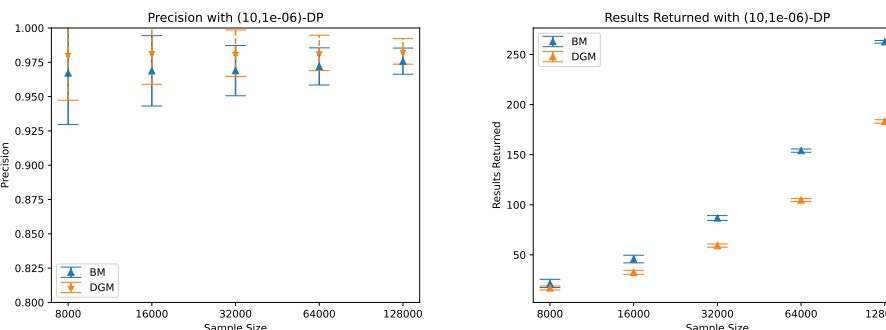

Figure 2: Precision and number of results returned on Zipfian data for various sample sizes.

We also evaluated our approach on real data from Reddit comments from `https://github.com/heyyjudes/differentially-private-set-union/tree/ea7b39285dace35cc9e9029692802759f3e1c8e8/data`. This data consists of comments from Reddit authors. To find the most frequent words from distinct authors, we take the set of all distinct words contributed by each author, which can be arbitrarily large and form the resulting histogram which has $\ell_\infty$-sensitivity 1 yet unbounded $\ell_2$-sensitivity. To get a domain of words to select from, we take the top-1000 words from this histogram. We note that this step should also be done with DP, but will not impact the relative performance between using the Brownian noise reduction and the Doubling Gaussian Method.

We then follow the same approach as on the synthetic data, using the Exponential Mechanism with $\varepsilon_{\text{EM}} = 0.01$, minimum privacy parameter $\varepsilon_n^{(1)} = 0.0001$, relative error $\alpha = 0.01$, and overall $(\varepsilon = 1, \delta = 10^{-6})$-DP guarantee. In 1000 trials, the Brownian noise reduction precision (proportion of results that had noisy counts within 1% of the true count) was on average 97% (with minimum 92%) while the Doubling Gaussian Method precision was on average 98% (with minimum 93%). Furthermore, the number of results returned by the Brownian noise reduction in 1000 trials was on average 152 (with minimum 151), while the number of results returned by the Doubling Gaussian method was on average 109 (with minimum 108).

# 8 Conclusion

We have presented a way to combine approximate zCDP mechanisms and ex-post private mechanisms while achieving an overall differential privacy guarantee, allowing for more general and flexible types of interactions between an analyst and a sensitive dataset. Furthermore, we showed how this type of composition can be used to provide overall privacy guarantees subject to outcomes satisfying strict accuracy requirements, like relative error. We hope that that this will help extend the practicality of private data analysis by allowing the release of counts with relative error bounds subject to an overall privacy bound.

## 9  Acknowledgements

We would like to thank Adrian Rivera Cardoso and Saikrishna Badrinarayanan for helpful comments. This work was done while G.S. was a visiting scholar at LinkedIn. ZSW was supported in part by the NSF awards 2120667 and 2232693 and a Cisco Research Grant.

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

# A  Concurrent Composition

We will use *concurrent composition* for differential privacy, introduced by Vadhan and Wang [2021] and with subsequent work in Lyu [2022], Vadhan and Zhang [2023], in our analysis, which we define next. We first define an interactive system.

**Definition A.1.** *An interactive system is a randomized algorithm $S : (\mathcal{Q} \times \mathcal{Y})^* \times \mathcal{Q} \to \mathcal{Y}$ with input an interactive history $(q_1, y_1), (q_2, y_2), \cdots, (q_t, y_t) \in (\mathcal{Q} \times \mathcal{Y})^t$ with a query $q_{t+1} \in \mathcal{Q}$. The output of $S$ is denoted $y_{t+1} \sim S((q_i, y_i)_{i \in [t]}, q_{t+1})$.*

Note that an interactive system may also consist of an input dataset $x$ that the queries are evaluated on, which will then induce an interactive system. In particular, we will consider two neighboring datasets $x^{(0)}$ and $x^{(1)}$, which will then induce interactive systems $S_i^{(b)}$ corresponding to input data $x^{(b)}$ for $b \in \{0, 1\}$ and $i \in [k]$. We will use the concurrent composition definition from Lyu [2022].

**Definition A.2** (Concurrent Composition). *Suppose $S_1, \cdots, S_k$ are $k$ interactive systems. The concurrent composition of them is an interactive system $\mathrm{COMP}(S_1, \cdots, S_k)$ with query domain $[k] \times \mathcal{Q}$ and response domain $\mathcal{Y}$. An adversary is a (possibly randomized) query algorithm $\mathcal{A} : ([k] \times \mathcal{Q} \times \mathcal{Y})^* \to [k] \times \mathcal{Q}$. The interaction between $\mathcal{A}$ and $\mathrm{COMP}(S_1, \cdots, S_k)$ is a stochastic process that runs as follows. $\mathcal{A}$ first computes a pair $(i_1, q_1) \in [k] \times \mathcal{Q}$, sends a query $q_1$ to $S_{i_1}$ and gets the response $y_1$. In the $t$-th step, $\mathcal{A}$ calculates the next pair $(i_t, q_t)$ based on the history, sends the $t$-th query $q_t$ to $S_{i_t}$ and receives $y_t$. There is no communication or interaction between the interactive systems. Each system $S_i$ can only see its own interaction with $\mathcal{A}$. Let $\mathrm{IT}(\mathcal{A} : S_1, \cdots, S_k)$ denote the random variable recording the transcript of the interaction.*

We will be interested in how much the distribution of the transcript of interaction changes for interactive systems with neighboring inputs. We will say $COMP(S_1, \cdots, S_k)$ is $(\varepsilon, \delta)$-DP if for all neighboring datasets $x^{(0)}, x^{(1)}$, we have for any outcome set of transcripts $E$, we have for $b \in \{0, 1\}$

$$\Pr[\mathrm{IT}(\mathcal{A} : S_1^{(b)}, \cdots, S_k^{(b)}) \in E] \le e^\varepsilon \Pr[\mathrm{IT}(\mathcal{A} : S_1^{(1-b)}, \cdots, S_k^{(1-b)}) \in E] + \delta.$$

We then have the following result from Lyu [2022].

**Theorem 6.** *Let $A_1, \cdots, A_k$ be $k$ interactive mechanisms that run on the same data set. Suppose that each mechanism $A_i$ satisfies $(\varepsilon_i, \delta_i)$-DP. Then $\mathrm{COMP}(\mathcal{A} : A_1, \cdots, A_k)$ is $(\varepsilon, \delta)$-DP, where $\varepsilon, \delta$ are given by the optimal (sequential) composition theorem Kairouz et al. [2017], Murtagh and Vadhan [2016].*

This result is very powerful because we can consider the privacy of each interactive system separately and still be able to provide a differential privacy guarantee for the more complex setting that allows an analyst to interweave queries to different interactive systems.

# B  Missing Proofs and Figures

## B.1  Missing Proofs from Section 5

We start with a simple result that states that ex-post mechanisms compose by just adding up the ex-post privacy functions. Similar to our notation for privacy filters, we will denote $\mathcal{E}_n(\cdot\,; x)$ as the privacy loss bound for algorithm $A_n$ conditioned on the outcomes of $A_{1:n-1}(x)$.

**Lemma B.1.** *Fix a sequence $\delta_1, \cdots, \delta_m \ge 0$. Let there be a probability measure $\mu_n$ on $\mathcal{Y}_n$ for each $n$ and the product measure on $\mathcal{Y}_1 \times \cdots \times \mathcal{Y}_m$. Consider mechanisms $A_n : \mathcal{X} \times \prod_{i=1}^{n-1} \mathcal{Y}_i \to \mathcal{Y}_n$ for $n \in [m]$ where each $A_n(\cdot\,; y_{1:n-1})$ is $(\mathcal{E}_n(\cdot\,; y_{1:n-1}), \delta_n)$-ex-post private for all prior outcomes $y_{1:n-1}$. Then the overall mechanism $A_{1:m}(\cdot)$ is $(\sum_{i=1}^m \mathcal{E}_i(A_i(\cdot)\,; \cdot), \sum_{i=1}^m \delta_i)$-ex-post private with respect to the product measure.*

*Proof.* We consider neighboring inputs $x, x'$ and write the privacy loss random variable $\mathcal{L}_A(A(x))$ for $A$ in terms of the privacy losses $\mathcal{L}_i(A_i(x))$ of each $A_i$ for $i \in [m]$

$$\Pr\left[\mathcal{L}_A(A(x)) < \sum_{i=1}^m \mathcal{E}_i(A_i(x); x)\right] = \Pr\left[\sum_{i=1}^m \mathcal{L}_i(A_i(x)) < \sum_{i=1}^m \mathcal{E}_i(A_i(x); x)\right]$$

Because each mechanism $A_i$ is $(\mathcal{E}_i(\cdot \ ; x), \delta_i)$-ex post private, we have $\Pr\left[\mathcal{L}_i(A(x)) \geq \mathcal{E}_i(A_i(x); x)\right] \leq \delta_i$ and hence

$$\Pr\left[\sum_{i=1}^{m} \mathcal{L}_i(A_i(x)) \leq \sum_{i=1}^{m} \mathcal{E}_i(A_i(x); x)\right] \geq \Pr\left[\bigcap_{i \in [m]} \{\mathcal{L}_i(A_i(x)) \leq \mathcal{E}_i(A_i(x); x)\}\right]$$

$$= 1 - \Pr\left[\bigcup_{i \in [m]} \{\mathcal{L}_i(A_i(x)) > \mathcal{E}_i(A_i(x); x)\}\right]$$

$$\geq 1 - \sum_{i \in [m]} \Pr\left[\mathcal{L}_i(A_i(x)) > \mathcal{E}_i(A_i(x); x)\right]$$

$$\geq 1 - \sum_{i \in [m]} \delta_i.$$

Hence, we have $\Pr\left[\mathcal{L}_A(A(x)) \geq \sum_{i=1}^{m} \mathcal{E}_i(A_i(x); x)\right] \leq \sum_{i \in [m]} \delta_i$, as desired. $\qquad \square$

For general ex-post private mechanisms, this *basic* composition cannot be improved. We can simply pick $\mathcal{E}_i$ to be the same as the privacy loss $\mathcal{L}_i$ at each round with independently selected mechanisms $A_i$ at each round $i \in [m]$. We now show how we can obtain a privacy filter from a sequence of ex-post mechanisms as long as each selected ex-post privacy mechanism selected at each round cannot exceed the remaining privacy budget. We will write $\delta_n(x)$ to denote the parameter $\delta_n$ selected as a function of prior outcomes from $A_1(x), \cdots, A_{n-1}(x)$.

**Lemma B.2.** *Let $\varepsilon > 0$ and $\delta \geq 0$ be fixed privacy parameters. Let $(A_n : \mathcal{X} \to \mathcal{Y})_{n \geq 1}$ be a sequence of $(\mathcal{E}_n(\cdot \ ; x), \delta_n(x))$-ex-post private conditioned on prior outcomes $y_{1:n-1} = A_{1:n-1}(x)$ where for all $y_i$ we have*

$$\sum_{i=1}^{n-1} \mathcal{E}_i(y_i; x) \leq \varepsilon.$$

*Consider the function $N : \mathcal{Y}^{\infty} \to \mathbb{N}$ where*

$$N((y_n)_{n \geq 1}) = \inf\left\{n : \varepsilon = \sum_{i \in [n]} \mathcal{E}_i(y_i; x) \ \text{ or } \ \delta < \sum_{i \in [n+1]} \delta_i(y_{1:i-1})\right\}.$$

*Then the algorithm $A_{1:N(\cdot)}(\cdot)$ is $(\varepsilon, \delta)$-DP where*

$$N(x) = N((A_n(x))_{n \geq 1}).$$

*Proof.* We follow a similar analysis in Whitehouse et al. [2023] where they created a filter for *probabilistic* DP mechanisms, that is the privacy loss of each mechanism can be bounded with high probability. We will write the corresponding privacy loss variables of $A_n$ to be $\mathcal{L}_n$ and for the full sequence of algorithms $A_{1:n} = (A_1, \cdots, A_n)$, the privacy loss is denoted as $\mathcal{L}_{1:n}$.

Define the events

$$D := \{\exists n \leq N(x) : \mathcal{L}_{1:n} > \varepsilon\}, \ \text{ and } \ E := \{\exists n \leq N(x) : \mathcal{L}_n > \mathcal{E}_n(A_n(x); x)\}.$$

Using Bayes rule, we have that

$$\Pr[D] = \Pr[D \cap E^c] + \Pr[D \cap E] \leq \Pr[D | E^c] + \Pr[E] = \Pr[E],$$

where the last inequality follows from how we defined the stopping function $N(x)$. Now, we show that $\Pr[E] \leq \delta$. Define the modified privacy loss random variables $(\widetilde{\mathcal{L}}_n)_{n \in \mathbb{N}}$ by

$$\widetilde{\mathcal{L}}_n := \begin{cases} \mathcal{L}_n & n \leq N(x) \\ 0 & \text{otherwise} \end{cases}.$$

Likewise, define the modified privacy parameter random variables $\widetilde{\mathcal{E}}_n(\cdot; x)$ and $\widetilde{\delta}_n(x)$ in an identical manner. Then, we can bound $\Pr[E]$ in the following manner:

$$\Pr[\exists n \leq N(x) : \mathcal{L}_n > \mathcal{E}_n(A_n(x); x)] = \Pr\left[\exists n \in \mathbb{N} : \widetilde{\mathcal{L}}_n > \widetilde{\mathcal{E}}_n(A_n(x); x)\right]$$

$$\leq \sum_{n=1}^{\infty} \Pr\left[\widetilde{\mathcal{L}}_n > \widetilde{\mathcal{E}}_n(A_n(x); x)\right]$$

$$= \sum_{n=1}^{\infty} \mathbb{E}\left[\Pr\left[\widetilde{\mathcal{L}}_n > \widetilde{\mathcal{E}}_n(A_n(x); x) \mid \mathcal{F}_{n-1}\right]\right]$$

$$\leq \sum_{n=1}^{\infty} \mathbb{E}\left[\widetilde{\delta}_n(x)\right] = \mathbb{E}\left[\sum_{n=1}^{\infty} \widetilde{\delta}_n(x)\right] = \mathbb{E}\left[\sum_{n \leq N(x)} \delta_n(x)\right] \leq \delta,$$

as desired. $\qquad\square$

*Proof of Theorem 2.* We will separate out the approximate zCDP mechanisms, $(A_n^{\mathrm{zCDP}})_{n \geq 1}$ from the ex-post private mechanisms $(A_n^{\mathrm{post}})_{n \geq 1}$ to form two separate interactive systems. In this case, the parameters that are selected can only depend on the outcomes from the respective interactive system, e.g. $\rho_n(x)$ can only depend on prior outcomes to mechanisms $A_i^{\mathrm{zCDP}}(x)$ for $i < n$. From Theorem 1, we know that $A_{1:N_{\mathrm{zCDP}}(\cdot)}^{\mathrm{zCDP}}(\cdot)$ is $(\varepsilon, \delta + \delta'')$-DP. We denote $\mathcal{L}_n^{\mathrm{post}}$ to be the privacy loss random variable for the ex-post private mechanism at round $n$. We will also write the stopping time for the ex-post private mechanisms as $N_{\mathrm{post}}(x)$. From Lemma B.2, we know that $A_{1:N_{\mathrm{post}}(\cdot)}^{\mathrm{post}}(\cdot)$ is $(\varepsilon', \delta')$-DP.

From Theorem 6, we know that the concurrent composition, which allows for both $A_{1:N_{\mathrm{post}}(\cdot)}^{\mathrm{post}}(\cdot)$ and $A_{1:N_{\mathrm{zCDP}}(\cdot)}^{\mathrm{zCDP}}(\cdot)$ to interact arbitrarily, will still be $(\varepsilon + \varepsilon', \delta + \delta' + \delta'')$-DP. $\qquad\square$

**Remark 1.** *Note that although DP is closed under post-processing, ex-post privacy is not. We typically say that any post processing function of a DP outcome is also DP with the same parameter. Our privacy analysis depends on getting actual outcomes from BM, rather than a post-processed value of it. However, we also say post processing ensures that no one can compromise privacy by taking the outcome and trying to learn more about the dataset. With ex-post privacy, we are not trying to ensure privacy of each outcome, but rather the full interaction, so the full interaction will still be DP under any post processing of any intermediate outcomes.*

### B.2 Missing Remarks from Section 6

We now write out some remarks regarding Theorem 4.

**Remark 2.** *We note here that the stopping time of each Brownian noise reduction is with respect to $\mathcal{H} = (\mathcal{H}^{(t)})$ where $\mathcal{H}^{(t)} = \sigma(f(x) + B^{(u)}; u > t)$, but $f(x)$ is constant so the shifted process is equivalent to the of the non-shifted process $\mathcal{G}$ as defined above.*

**Remark 3.** *For the multivariate case we have $B^{(u)} = \left(B^{(u)}[i] : i \in [d]\right)$ where each coordinate is an independent Brownian motion, and we will write the filtration $\mathcal{G}[i]$ to be the natural reverse filtration corresponding to the Brownian motion for index $i$. We then define $\mathcal{G}^{(t)} = \sigma\left(B^{(u)}[1], \cdots, B^{(u)}[d] : u \geq t\right)$. From Lemma 6.1, we have the following for all $\lambda \in \mathbb{R}$, $i \in [d]$, and $0 < s < t$*

$$\mathbb{E}[\exp(\lambda B^{(t)}[i]/t - \lambda^2/(2t)) \mid \mathcal{G}^{(s)}[i]] \leq 1.$$

We then consider the full d-dimensional Brownian motion so that with a unit vector $v = (v[1], \cdots, v[d]) \in \mathbb{R}^d$ we have

$$\mathbb{E}\left[\exp\left(\lambda \cdot \sum_{i=1}^{d} v[i] \cdot B^{(t)}[i]/t - \frac{\lambda^2}{2t}\right) \mid \mathcal{G}^{(s)}\right]$$

$$= \mathbb{E}\left[\prod_{i=1}^{d} \exp\left(\lambda \cdot v[i] \cdot B^{(t)}[i]/t - \frac{\lambda^2 \cdot v[i]^2}{2t}\right) \mid \mathcal{G}^{(s)}\right]$$

$$= \prod_{i=1}^{d} \mathbb{E}\left[\exp\left(\lambda \cdot v[i] \cdot B^{(t)}[i]/t - \frac{\lambda^2 \cdot v[i]^2}{2t}\right) \mid \mathcal{G}^{(s)}[i]\right]$$

$$\leq 1.$$

## B.3 Missing Proofs from Section 6

*Proof of Lemma 6.2.* We use the decomposition of the privacy loss $\mathcal{L}_n^{(1:k)}$ at round $n \geq 1$ for Brownian noise reduction in Theorem 3 that is stopped at time value $t_n^{(k)}$ to get the following with the natural filtration $(\mathcal{F}_n(x))_{n \geq 1}$ generated by $(A_n(x))_{n \geq 1}$. Recall that we have for the Brownian motion $(B_n^{(t)})_{t > 0}$ used in the Brownian noise reduction

$$\mathcal{L}_n^{(1:k)} = \mathcal{L}_n^{(k)} = \frac{1}{2t_n^{(k)}} + \frac{B_n^{(t_n^{(k)})}}{t_n^{(k)}}.$$

From Lemma 6.1 we have for all $\lambda \in \mathbb{R}$ and $n \geq 1$ with time value $t_n^{(k)}$

$$\mathbb{E}\left[\exp\left(\lambda X_n^{(k)} - \frac{\lambda^2}{2t_n^{(k)}}\right) \mid \mathcal{F}_{n-1}(x)\right] \leq 1, \qquad \text{where} \qquad X_n^{(k)} = \mathcal{L}_n^{(k)} - \frac{1}{2t_n^{(k)}}.$$

We then form the following process

$$M_n^{(k_1, \cdots, k_n)} = \exp\left(\lambda \sum_{i=1}^{n} X_i^{(k_i)} - \sum_{i=1}^{n} \frac{\lambda^2}{2t_i^{(k_i)}}\right).$$

Hence, with fixed time values $(t_n^{(k_n)})_{n \geq 1}$ for $n \geq 1$ we have for all $\lambda$,

$$\mathbb{E}\left[M_n^{(k_1, \cdots, k_n)} \mid \mathcal{F}_{n-1}(x)\right] \leq 1.$$

We then replace $(k_i)_{i \leq n}$ with an adaptive stopping time functions $(T_i)_{i \leq n}$, rather than fixing them in advance, in which case we rename $M_n^{(k_1, \cdots, k_n)}$ as $M_n^{\mathrm{BM}}$. We know from Lemma 6.1 that $M_n^{\mathrm{BM}}$ is still an e-value for any $n$. We then apply the optional stopping theorem to conclude that with the stopping time $N(x)$ that $\mathbb{E}\left[M_{N(x)}^{\mathrm{BM}}\right] \leq 1$. By the definition of our stopping time, so that $\sum_{i=1}^{N(x)} \frac{1}{2t_i^{T_i(x)}} = \rho$, we have for all $\lambda$,

$$\mathbb{E}\left[\exp\left((\lambda - 1) \sum_{i=1}^{N(x)} L_i^{(T_i(x))}\right)\right] \leq e^{\lambda(\lambda-1)\rho}.$$

We then set $\lambda = \frac{2\rho + 2\sqrt{\rho \log(1/\delta)}}{2\rho}$ to get

$$\Pr\left[\sum_{i=1}^{N(x)} L_i^{(T_i(x))} \geq \rho + 2\sqrt{\rho \log(1/\delta)}\right] \leq \delta.$$

We then have a high probability bound on the overall privacy loss, which then implies differential privacy. □

*Proof of Theorem 5.* We will show that $A_{1:N(\cdot)}(\cdot)$ is $\delta$-approximate $\rho$-zCDP and then use Lemma 2.1 to obtain a DP guarantee stopping at $N(x)$ as defined in the theorem statement.

We follow a similar analysis to the proof of Theorem 1 in Whitehouse et al. [2023]. Let $P_{1:n}$ and $Q_{1:n}$ denote the joint distributions of $(A_1, \ldots, A_n)$ with inputs $x$ and $x'$, respectively. We overload notation and write $P_{1:n}(y_1, \ldots, y_n)$ and $Q_{1:n}(y_1, \ldots, y_n)$ for the likelihood of $y_1, \ldots, y_n$ under input $x$ and $x'$ respectively. We similarly write $P_n(y_n \mid y_{1:n-1})$ and $Q_n(y_n \mid y_{1:n-1})$ for the corresponding conditional densities.

For any $n \in \mathbb{N}$, we have

$$P_{1:n}(y_1, \cdots, y_n) = \prod_{m=1}^{n} P_m(y_m \mid y_{1:m-1}),$$

$$Q_{1:n}(y_1, \cdots, y_n) = \prod_{m=1}^{n} Q_m(y_m \mid y_{1:m-1}).$$

When then show that the two likelihoods can be decomposed as weighted mixtures of $P'$ and $P''$, as well as $Q'$ and $Q''$, respectively such that the mixture weights on $P'$ and $Q'$ are at least $(1 - \delta)$, and for all $\lambda \geq 1$,

$$\max\left\{ D_\lambda \left(P' \| Q'\right), D_\lambda \left(Q' \| P'\right) \right\} \leq \rho\lambda. \tag{4}$$

By our assumption of approximate zCDP at each step $n$, we can write the conditional likelihoods of $P_n$ and $Q_n$ as the following convex combinations:

$$P_n(y_n \mid y_{1:n-1}) = (1 - \delta_n(y_{1:n-1}))P'_n(y_n \mid y_{1:n-1}) + \delta_n(y_{1:n-1})P''_n(y_n \mid y_{1:n-1}),$$

$$Q_n(y_n \mid y_{1:n-1}) = (1 - \delta_n(y_{1:n-1}))Q'_n(y_n \mid y_{1:n-1}) + \delta_n(y_{1:n-1})Q''_n(y_n \mid y_{1:n-1}),$$

such that for all $\lambda \geq 1$ and all prior outcomes $y_{1:n-1}$, we have both

$$D_\lambda \left(P'_n(\cdot \mid y_{1:n-1}) \| Q'_n(\cdot \mid y_{1:n-1})\right) \leq \rho_n(y_{1:n-1})\lambda, \tag{5}$$

$$D_\lambda \left(Q'_n(\cdot \mid y_{1:n-1}) \| P'_n(\cdot \mid y_{1:n-1})\right) \leq \rho_n(y_{1:n-1})\lambda. \tag{6}$$

Note that at each round, we either select an approximate zCDP mechanism or select a Brownian noise reduction, and in the latter case $\delta_n(x) \equiv 0$ and $\rho_n(x) \equiv 0$, which then means $P'_n \equiv P_n$ and $Q'_n \equiv Q_n$ at those rounds $n$. We will write the distribution $P_{1:n}$ for any prefix of outcomes from $A_1(x), \cdots, A_n(x)$ and similarly we will write the distribution $Q_{1:n}$ for the prefix of outcomes from $A_1(x'), \cdots, A_n(x')$. We can then write these likelihood as a convex combination of likelihoods, using the fact that $\sum_{n=1}^{\infty} \delta_n(y_{1:n-1}) \leq \delta$ for all $y_1, y_2, \cdots$.

$$P_{1:n}(y_1, \cdots, y_n) = (1 - \delta)\underbrace{\prod_{\ell=1}^{n} P'_\ell(y_\ell \mid y_{1:\ell-1})}_{P'_{1:n}(y_1, \cdots, y_n)} + \delta P''_{1:n}(y_1, \cdots, y_n). \tag{7}$$

$$Q_{1:n}(y_1, \cdots, y_n) = (1 - \delta)\underbrace{\prod_{\ell=1}^{n} Q'_\ell(y_\ell \mid y_{1:\ell-1})}_{Q'_{1:n}(y_1, \cdots, y_n)} + \delta Q''_{1:n}(y_1, \cdots, y_n). \tag{8}$$

For any fixed $\lambda \geq 1$ and $k \geq 1$, consider the following filtration $\mathcal{F}' = (\mathcal{F}'_n)_{n \geq 1}$ where $\mathcal{F}'_n = \sigma(Y'_1, \cdots, Y'_n)$, with $Y'_1 \sim P'_1, Y'_2 \mid Y'_1 \sim P'_2(\cdot \mid Y'_1), \cdots, Y'_k \mid Y'_{1:k-1} \sim P'_k(\cdot \mid Y'_{1:k-1})$.

We will first consider the Brownian noise reduction mechanisms with time values $(t_n^{(k)} : k \geq 1)_{n \geq 1}$ to be stopped at fixed rounds $(k_n)_{n \geq 1}$, although not every round will have a Brownian noise reduction mechanism selected with Brownian motion $(B_n^{(t)})_{t > 0}$. We will write out the privacy loss for the Brownian noise reduction mechanisms stopped at rounds $(k_n)_{n \geq 1}$ as $\mathcal{L}_n^{(1:k_n)}$ where from Theorem 3 we have,

$$\mathcal{L}_n^{(1:k_n)} = \mathcal{L}_n^{(k_n)} = \frac{1}{2t_n^{(k_n)}} + \frac{B_n^{(k_n)}}{t_n^{(k_n)}}.$$

We will add the noise reduction outcomes to the filtration, so that $\mathcal{F}_n'' = \sigma(Y_1', \cdots, Y_{n-1}', (B_n^{(t)})_{t \geq k_n})$. From Lemma 6.1, we know for all $\lambda \in \mathbb{R}$

$$\mathbb{E}\left[\exp\left(\lambda\left(\mathcal{L}_n^{(1:k_n)} - \frac{1}{2t_n^{(k_n)}}\right) - \frac{\lambda^2}{2t_n^{(k_n)}}\right) \mid \mathcal{F}_{n-1}''\right]$$
$$= \mathbb{E}\left[\exp\left(\lambda\mathcal{L}_n^{(1:k_n)} - \frac{\lambda(\lambda+1)}{2t_n^{(k_n)}}\right) \mid \mathcal{F}_{n-1}''\right]$$
$$\leq 1.$$

We then replace $(k_i)_{i \leq n}$ with an adaptive stopping time functions $(T_i)_{i \leq n}$ with corresponding stopping times $(T_i(x))_{i \leq n}$, rather than fixing them in advance, in which case we rename $\mathcal{L}_n^{(1:k_n)}$ as $\mathcal{L}_n^{\text{BM}_n}$ and the same inequality still holds with the filtration $\mathcal{F}_n''' = \sigma(Y_1', \cdots, Y_{n-1}', (B_n^{(t)})_{t \geq T_n(x)})$. Note that we will call $Y_n' = (f(x) + B_n^{(t)})_{t \geq T_n(x)}$ so that $\mathcal{F}_n''' = \mathcal{F}_n'$.

At rounds $n$ where a Brownian noise reduction mechanism is not selected, we simply have $1/t_n^{(k_n)} = 0$ and $\mathcal{L}_n^{(1:k_n)} = 0$. We also consider the modified privacy losses for the approximate zCDP mechanisms $\mathcal{L}_n'$ where

$$\mathcal{L}_n' = \mathcal{L}_n'(y_{1:n}) = \log\left(\frac{P_n'(y_n|y_{<n})}{Q_n'(y_n|y_{<n})}\right), \quad \text{where } y_{1:n} \sim P_{1:n}'.$$

Due to mechanisms being zCDP, we then have for any $\lambda \geq 1$

$$\mathbb{E}\left[\exp\left((\lambda-1)\mathcal{L}_n' - \lambda(\lambda-1)\rho_n(Y_{1:n-1}')/2\right) \mid \mathcal{F}_{n-1}'\right] \leq 1.$$

Because at each round $n$, the mechanism selected is either approximate zCDP or a Brownian noise reduction with a stopping function, we can write the privacy loss at each round $i$ as the sum $\mathcal{L}_i' + \mathcal{L}_i^{\text{BM}}$ so that for all $\lambda \geq 1$ we have

$$X_n^{(\lambda)} := \sum_{i \leq n} \left\{ \mathcal{L}_i' + \mathcal{L}_i^{\text{BM}} - \lambda\left(\rho_i(Y_{1:n-1}') + \frac{1}{t_i^{(T_i(x))}}\right) \right\}, \tag{9}$$

$$M_n^{(\lambda)} := \exp\left((\lambda-1)X_n^{(\lambda)}\right). \tag{10}$$

We know that $(M_n)$ is a supermartingale with respect to $(\mathcal{F}_n')_{n \geq 1}$. From the optional stopping theorem, we have

$$\mathbb{E}[M_{N(x)}^{(\lambda)}] \leq 1.$$

This will ensure (4) holds. Although for rounds $n$ where we select a Brownian noise reduction we have $P_n' = P_n$ and $Q_n' = Q_n$, we still need to show that for rounds $n$ where approximate zCDP mechanisms were selected the original distributions $P_n$ and $Q_n$ can be written as weighted mixtures including $P_n'$ and $Q_n'$, respectively. This follows from the same analysis as in Whitehouse et al. [2023], so that for all outcomes $y_1, y_2, \cdots$ where $\sum_{n=1}^{\infty} \delta_n(y_{1:n-1}) \leq \delta$ we have

$$P_{1:n}(y_1, y_2, \cdots, y_n) \geq (1-\delta) \prod_{m=1}^{n} P_m'(y_m \mid y_{1:m-1}),$$

and similarly for $Q_{1:n}$. As is argued in Whitehouse et al. [2023], it suffices to show that the two likelihoods of the stopped process $P_{1:N}$ and $Q_{1:N}$ can be decomposed as weighted mixtures of $P_{1:N}'$ and $P_{1:N}''$ as well as $Q_{1:N}'$ and $Q_{1:N}''$, respectively such that the weights on $P_{1:N}'$ and $Q_{1:N}'$ are at least $1 - \delta$. Note that from our stopping rule, we haven for all $\lambda > 0$

$$\mathbb{E}[M_{N(x)}^{(\lambda)}] \leq 1 \implies \mathbb{E}\left[e^{\lambda \sum_{n=1}^{N(x)}(\mathcal{L}_n' + \mathcal{L}_n^{\text{BM}})}\right] \leq e^{\lambda(\lambda+1)\rho}.$$

We then still need to convert to DP, which we do with the stopping rule $N(x)$ and the conversion lemma between approximate zCDP and DP in Lemma 2.1.

$\square$

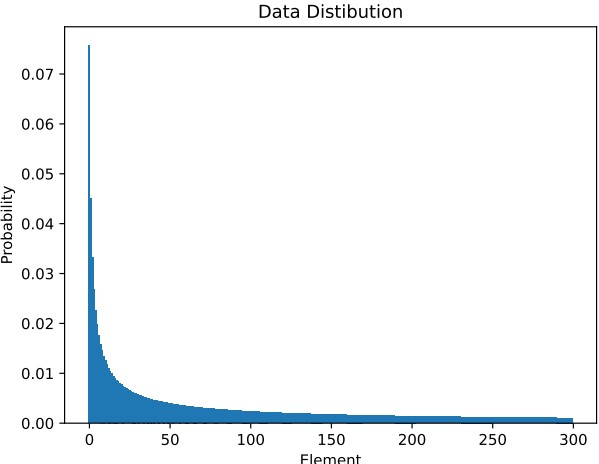

Figure 3: Data from a Zipf Distribution with parameter $a = 0.75$ and max value of 300.

## B.4 Data Distribution Plot from Section 7

In Section 7 we generate data from a Zipfian distribution, which we present in Figure 3.

## C Noise Reduction Mechanisms

We provide a more general approach to handle both the Laplace and Brownian noise reduction mechanisms from Koufogiannis et al. [2017], Ligett et al. [2017], Whitehouse et al. [2022]. Consider a discrete set of times $\{a^{(j)}\}$ with $0 < a^{(j)} \leq b$. We will allow the stopping time and the number of steps to be adaptive. We will handle the univariate case and note that the analysis extends to the multivariate case. We will write time functions $(\phi^{(k)}, k \geq 1)$ to satisfy the following

$\phi^{(1)} \equiv b > 0$,

$\phi^{(k)} : \{a^{(j)}\} \times \mathbb{R} \to \{a^{(j)}\}$ such that $\phi^{(k)}(t, z) < t$ for all $(t, z) \in \{a^{(j)}\} \times \mathbb{R}$ and $k \geq 2$.

Consider the noise reduction mechanisms where we have $M^{(1,p)} = t^{(1)} = b$, $M^{(1,q)}(x) = f(x) + Z(t^{(1)})$ where $Z(t)$ is either a standard Brownian motion or a Laplace process Koufogiannis et al. [2017], and for $k > 1$ we have

$$M^{(k,p)}(x) = \phi^{(k)}(M^{(k-1,p)}(x), M^{(k-1,q)}(x)),$$
$$M^{(k,q)}(x) = f(x) + Z(M^{(k,p)}(x)).$$

We will then write

$$M^{(1:k)}(x) = \left((M^{(k,p)}(x), M^{(k,q)}(x)), \cdots, (M^{(1,p)}(x), M^{(1,q)}(x))\right).$$

We then consider the extended mechanism $M^*(x) = (T(x), M^{(1:T(x))}(x))$, where $T(x)$ is a stopping function. Hence, $M^*(x)$ takes values in $\mathbb{Z} \times ((0, b) \times \mathbb{R})^{\infty,*}$ where $((0, b] \times \mathbb{R})^{\infty,*}$ is a collection of all finite sequences of the type $((t^{(i)}, z^{(i)}) : i \in [k])$ for $k = 1, 2, \cdots$ and $0 < t^{(k)} < \cdots, t^{(1)} = b$ and $z^{(i)} \in \mathbb{R}$ for $i \in [k]$.

Let $m^*$ be the probability measure on this space generated by $M^*(x^*)$ where $x^*$ is a point where $f(x^*) = 0$. Note that $x^*$ might be a fake value that needs to be added to make the function equal zero. Let $m(x)$ be the probability measure on this space generated by $M^*(x)$. We will then compute densities with respect to $m^*$.

Let $A$ be a measurable set in $\mathbb{Z} \times ((0, b] \times \mathbb{R})^{\infty,*}$. Let $A^{(j)} \subseteq A$ on which the last (smallest) $p$-coordinate of every point is equal to $a^{(j)}$. Then

$$\Pr[M^*(x) \in A] = \sum_j \Pr[M^*(x) \in A^{(j)}].$$

By construction, $M^*(x)$ is a function of $(Z(t) + f(x), 0 \le t \le b)$. Therefore, for each $j$, we have

$$\Pr[M^*(x) \in A^{(j)}] = \Pr[(Z(t) + f(x), 0 \le t \le b) \in (M^*)^{-1}(A^{(j)})],$$

where the set of paths is of the following form for a measurable subset $D^{(j)} \in C([a^{(j)}, b])$ of continuous functions on $[a^{(j)}, b]$:

$$(M^*)^{-1}(A^{(j)}) = \left\{ (y(t) : 0 \le t \le b) : (y(t) : a^{(j)} \le t \le b) \in D^{(j)} \right\}.$$

The standard Brownian motion and the Laplace process both have independence of increments, so we have the following with $p_t$ as the density for $Z(t)$

$$\Pr[(Z(t) + f(x) : 0 \le t \le b) \in \{(y(t) : 0 \le t \le b) : (y(t) : a^{(j)} \le t \le b) \in D^{(j)}\}]$$
$$= \Pr[(Z(t) + f(x) : a^{(j)} \le t \le b) \in D^{(j)}]$$
$$= \mathbb{E}\left[ \mathbb{1}\left\{ (Z(t) : a^{(j)} \le t \le b) \in D^{(j)} \right\} \frac{p_{a^{(j)}}(Z(a^{(j)}) - f(x))}{p_{a^{(j)}}(Z(a^{(j)}))} \right]$$
$$= \mathbb{E}\left[ \mathbb{1}\left\{ M^*(x^*) \in A^{(j)} \right\} \frac{p_{M^{(T(x^*),q)}(x^*)}(Z(M^{(T(x^*),q)}(x^*)) - f(x))}{p_{M^{(T(x^*),q)}(x^*)}(Z(M^{(T(x^*),q)}(x^*)))} \right],$$

where the second equality follows from the fact that the law of a shifted process with independent increments on $(a^{(j)}, b)$ is equivalent to the law of the non-shifted process and its density is the ratio of the two densities evaluated at its left most point. We then conclude that

$$\Pr[M^*(x) \in A] = \mathbb{E}\left[ \mathbb{1}\left\{ M^*(x^*) \in A \right\} \frac{p_{M^{(T(x^*),p)}(x^*)}(Z(M^{(T(x^*),p)}(x^*)) - f(x))}{p_{M^{(T(x^*),p)}(x^*)}(Z(M^{(T(x^*),p)}(x^*)))} \right].$$

Therefore $m(x)$ is absolutely continuous with respect to $m^*$. To write the density (the Radon-Nikodym derivative), recall that the density is evaluated at some point $\mathbf{s} \in \mathbb{Z} \times (\{a^{(j)}\} \times \mathbb{R})^{\infty,*}$. Let $t(\mathbf{s})$ be the smallest $p$-value in $\mathbf{s}$ and $z(\mathbf{s})$ be the corresponding space value. Then we have

$$p_x^{(1:T(x))}(\mathbf{s}) = \frac{dm(x)}{dm^*}(\mathbf{s}) = \frac{p_{t(\mathbf{s})}(z(\mathbf{s}) - f(x))}{p_{t(\mathbf{s})}(z(\mathbf{s}))}.$$

Similarly, we consider $M^*_{\text{last}}(x) = (T(x), M^{(T(x))}(x))$ so that the state space is $\mathbb{Z} \times \{a^{(j)}\} \times \mathbb{R}$. We will write $\hat{m}^*$ to be the probability measure on this space generated by $M^*_{\text{last}}(x^*)$ and $\hat{m}(x)$ to be the probability measure on this space generated by $M^*_{\text{last}}(x)$. We then compute densities with respect to $\hat{m}^*$.

Let $A$ now be a measurable subset of $\mathbb{Z} \times \{a^{(j)}\} \times \mathbb{R}$ and let $A^{(j)} \subseteq A$ with the $p$-coordinate equal to $a^{(j)}$. Then we follow a similar argument to what we have above to show that

$$\Pr[M^*_{\text{last}}(x) \in A] = \mathbb{E}\left[ \mathbb{1}\left\{ M^*_{\text{last}}(x^*) \in A \right\} \frac{p_{M^{(T(x^*),p)}(x^*)}(Z(M^{(T(x^*),p)}(x^*)) - f(x))}{p_{M^{(T(x^*),p)}(x^*)}(Z(M^{(T(x^*),p)}(x^*)))} \right].$$

Hence, for $(k, t, z) \in \mathbb{Z} \times \{a^{(j)}\} \times \mathbb{R}$ we have

$$p_x^{(T(x))}(k, t, z) = \frac{dm(x)}{dm^*}(k, t, z) = \frac{p_t(z - f(x))}{p_t(z)}.$$

We then consider the privacy loss for both $M^*$, denoted as $\mathcal{L}^{(1:T(x))}$, and $M^*_{\text{last}}$, denoted as $\mathcal{L}^{(T(x))}$, under neighboring data $x$ and $x'$. We have

$$\mathcal{L}^{(1:T(x))} = \log \left( \frac{p_{M^{(T(x),p)}(x)}(Z(M^{(T(x),p)}(x)) - f(x))}{p_{M^{(T(x),p)}(x)}(Z(M^{(T(x),p)}(x)) - f(x'))} \right) = \mathcal{L}^{(T(x))}. \tag{11}$$

We then instantiate the Brownian noise reduction, in which case

$$p_t(z - f(x))/p_t(z) = \exp\left(-\frac{1}{2t}(z - f(x))^2 + \frac{1}{2t}z^2\right) = \exp\left(\frac{f(x)}{t}z - \frac{f(x)^2}{2t}\right).$$

Without loss of generality, consider the function $g(y) = f(y) - f(x)$ so that we apply

$$\log\left(\frac{p_t(z - g(x))}{p_t(z - g(x'))}\right) = -\frac{g(x')}{|g(x')|t}z|g(x')| + \frac{g(x')^2}{2t}.$$

We then use the Brownian motion $(B(t))_{t \geq 0}$ to get the privacy loss $\mathcal{L}_{\text{BM}}^{(1:T(x))}$

$$
\begin{aligned}
\mathcal{L}_{\text{BM}}^{(1:T(x))} &= \frac{(f(x) - f(x')^2}{2t} - \frac{f(x) - f(x')}{|f(x) - f(x')|t}|f(x) - f(x')| \cdot B\left(M^{(T(x),p)}(x)\right) \\
&= \frac{(f(x) - f(x')^2}{2t} - \frac{1}{t}|f(x) - f(x')| \cdot W\left(M^{T(x),p}(x)\right),
\end{aligned}
$$

where $(W(t) = z \cdot B(t))$ is a standard Brownian motion for $|z| = 1$.

There is another noise reduction mechanism based on Laplace noise, originally from Koufogiannis et al. [2017] and shown to be ex-post private in Ligett et al. [2017]. We first show that it is indeed a noise reduction mechanism with a stopping function and consider the resulting privacy loss random variable. We focus on the univariate case, yet the analysis extends to the multivariate case.

We construct a Markov process $(X(t))_{t \geq 0}$ with $X(0) = 0$ such that, for each $t > 0$, $X(t)$ has the Laplace distribution with parameter $t$, which has density $p(z) = \frac{1}{2t}\exp\left(-|z|/t\right)$ for $z \in \mathbb{R}$. The process we construct has independent increments, i.e. for any $0 \leq t^{(1)} < t^{(2)} < \cdots < t^{(k)}$ the following differences are independent,

$$X(t^{(0)}), X(t^{(1)}) - X(t^{(0)}), \cdots, X(t^{(k)}) - X(t^{(k-1)}).$$

Hence, the process is Markovian. The idea of constructing such a process is that a Laplace random variable with parameter $t > 0$ is a symmetric, infinitely divisible random variable without a Gaussian component whose Lévy measure has density

$$g(z;t) = \frac{e^{-|z|/t}}{|z|}, \; z \neq 0.$$

Let $U$ be an infinitely divisible random measure on $[0, \infty)$ with Lebesgue control measure and local Lévy density

$$\psi(z;t) = t^{-2}e^{-|z|/t}, \; z \neq 0, t > 0. \tag{12}$$

We define $X(t) = U([0, t])$, where $t \geq 0$. Then $X(0) = 0$ and for $t > 0$, $X(t)$ is a symmetric, infinitely divisible random variable without a Gaussian component, whose Lévy measure has the density equal to

$$\int_0^t \psi(z;u)du = \int_0^t u^{-2}e^{-|z|/u}du = \int_{1/t}^\infty e^{-|z|s}ds = g(z;t).$$

That is $X(t)$ is a Laplace random variable with parameter $t$ and the resulting process has independent increments by construction. Note that the process $(X(t))_{t \geq 0}$ has infinitely many jumps near 0. On an interval $[t^{(1)}, t^{(2)}]$ with $0 < t^{(1)} < t^{(1)}$, it is flat with probability $(t^{(1)}/t^{(2)})^2$.

We can then describe the Laplace noise reduction algorithm in a way similar to the Brownian Noise Reduction.

**Definition C.1** (Laplace Noise Reduction). *Let $f : \mathcal{X} \to \mathbb{R}^d$ be a function and $(t^{(k)})_{k \geq 1}$ a sequence of time values. Let $(X(t))_{t \geq 0}$ be the Markov process described above independently in each coordinate and $T : (\mathbb{R}^d)^* \to \mathbb{N}$ be a stopping function. The Laplace noise reduction associated with $f$, time values $(t^{(k)})_{k \geq 1}$, and stopping function $T$ is the algorithm $\texttt{LNR} : \mathcal{X} \to ((0, t^{(1)}] \times \mathbb{R}^d)^*$ given by*

$$\texttt{LNR}(x) = \texttt{LNR}^{(1:T(x))}(x) = \left(t^{(k)}, f(x) + X(t^{(k)})\right)_{k \leq T(x)}.$$

We then aim to prove the following

**Lemma C.1.** *The privacy loss $\mathcal{L}_{LNR}$ for the Laplace noise reduction associated with $f$, time values $(t^{(k)})_{k \geq 1}$, and stopping function $T$ can be written as*

$$\mathcal{L}_{LNR} = \mathcal{L}_{LNR}^{(1:T(x))} = \mathcal{L}_{LNR}^{(T(x))}.$$

*Furthermore we have*

$$\mathcal{L}_{LNR}^{(1:T(x))} = -\frac{1}{M^{(T(x),p)}(x)}\left(|X(M^{(T(x),p)}(x))| - |X(M^{(T(x),p)}(x)) - \Delta_1(f)|\right).$$

*Proof.* Because the Laplace process has independent increments, we get the same expression for the privacy loss as in (11) where we substitute the Laplace process for $Z(t)$, which has the following density

$$p_t(z - f(x))/p_t(z) = \exp\left(-\frac{1}{t}\left(|z - f(x)| - |z|\right)\right).$$

We will again use $g(y) = f(y) - f(x)$ for neighboring datasets $x, x'$ with $f(x') > f(x)$ without loss of generality to get

$$\log\left(p_t(z - g(x))/p_t(z - g(x'))\right) = -\frac{1}{t}\left(|z| - |z - g(x')|\right).$$

Therefore, we have from (11) that

$$\begin{aligned}\mathcal{L}_{LNR}^{(1:T(x))} &= \mathcal{L}_{LNR}^{(T(x))}\\ &= -\frac{1}{M^{(T(x),p)}(x)}\left(|X(M^{(T(x),p)}(x))| - |X(M^{(T(x),p)}(x)) - |f(x) - f(x')||\right),\end{aligned}$$

as desired. $\qquad\square$

We now want to show a similar result that we had for the Brownian noise reduction mechanism in Lemma 6.1, but for the Laplace process. Instead of allowing general time step functions $\phi^{(j)}$, we will simply look at time values $0 < t^{(k)} < \cdots < t^{(1)} = b$ so that

$$\mathcal{L}_{LNR}^{(1:T(x))} = -\frac{1}{t^{(T(x))}}\left(|X(t^{(T(x))})| - |X(t^{(T(x))}) - |f(x) - f(x')||\right).$$

This form of the privacy loss for the Laplace noise reduction might be helpful in determining whether we can get a similar backward martingale as in the Brownian noise reduction case in Lemma 6.1. We leave the problem open to try to get a composition bound for Laplace noise reduction mechanisms that improves over simply adding up the ex-post privacy bounds as in Theorem B.1.

