# OpenReview forum: "Adaptive Privacy Composition for Accuracy-first Mechanisms"
_NeurIPS.cc/2023/Conference — NeurIPS 2023 poster_

### Official Review · Reviewer_dJHg · 2023-07-03

**Soundness:** 3 good
**Presentation:** 4 excellent
**Contribution:** 3 good
**Rating:** 6
**Confidence:** 3

**Summary:**

This paper presents extensive background on ex-post-privacy as well as some new analysis providing a new method for leveraging this framework for the design of differentially private interactive protocols. Finally, this framework is instantiated in a method for releasing private counts while targeting an accuracy constraint.

**Strengths:**

* The theoretical framework presented is quite general and seems to be very strong.
* Though relaxed notions of privacy are used internally in mechanism design, the main results here are stated for more tradition $(\epsilon, \delta)$ DP.
* The power of the theoretical framework _could_ conceviably lead to quite substantial concrete implementations and applications.

**Weaknesses:**

* The generality of the analysis yields an associated weakness: for the vast majority of the main body, the precise methods of application are somewhat unclear. E.g. it is not obvious until the end of the paper that we will not literally be guaranteeing that all of our released values are withing percent, which makes the design of interactive protocols which use this style of mechanism + composition an important and more or less unstated problem space.

* The point above can be restated in more direct terms: if the authors wish to see this analysis and framework have a big impact, it is probably still to much of a lift to ask the community to fill in the gaps between the abstract presentation which dominates here and concrete algorithmic implementations.


**Questions:**

* IIUC, the final guarantee claimed here is epsilon, delta DP. Is this correct? If so, I think the paper could benefit from a stronger statement of this much earlier. E.g. the abstract uses ‘differentially private’ and ‘ex post private’, but the statement about an overall guarantee just uses the word ‘privacy’. This leaves a reader somewhat confused about which notion of privacy applies here.

* In theorem 3, what _exactly_ do we mean by statistic and its sensitivity? IE, a Brownian motion itself has no data-dependence, but a Brownian noise reduction mechanism does. Is the notion of statistic and sensitivity we are using here something like: the Brownian noise-reduction mechanism is data-dependent, so its path is data-dependent, and the statistic is a function of the path--so for every neighboring $x, x'$, the function $f_i$ of these different paths can differ by no more than 1, _uniformly_ over the randomness in the BM? This would seem to me to be a somewhat different notion than that of usual sensitivity, where there is no randomness involved. Perhaps the term 'statistic' is intended to imply that the randomness has already been integrated out? But if so, what does this mean precisely (e.g. what would an example be)? Further, it seems like the result of Theorem 3 is independent of these statistics $f_i$, unless they are hiding in the stopping times $T_i$ somewhere? NOTE: it is possible that this $f_i$ is intended to recall $f$ of definition 4.4; but if so, I am not quite grokking this relationship--since the $f$ in definition 4.4 can, I guess, be viewed as one _particular_ statistic of the function $BM(x)$ (the limit of the expectation of the second toordinate as $t \to 0$), but doesn't really provide any immediate pointers to generalizing this notion?

* It's not necessarily clear from the presentation of the paper what relationship there is between ex-post mechanisms, noise-reduction mechanisms, and the Brownian noise reduction mechanism. Meaning, e.g. the statement in definition 4.2 seems to apply to the privacy loss random variable directly, considered as a function of both its inputs and its randomness--at least, this is how I interpret the equality in definition 2, equality of functions. If this is true, seemingly this definition would be applicable to any particular property of the functions $L$; e.g. if $\cal{L}^{(k)}$ was $(\epsilon, \delta)$-DP then $\cal{L}^{(1:k)}$ would be as well (since it is just a statement about the function $L$, and these functions are equal). But thinking about the domains and ranges, it seems like these functions take values in different spaces. So it's not clear, I suppose, what this equality means? Maybe there is a suppressed relationship to ex-post privacy?

* If I read correctly the definition of $(\cal{E}, \delta, \epsilon^*)$ ex-post-private (internal to definition 2) is equivalent to $(\epsilon^*, \delta)$ DP. Is this correct? If so, readers may find it useful to have this explicitly called out, since likely many are more familiar with DP than ex-post privacy.

* Nit on eqn (4): IIUC, choosing the particular functional form here (e.g. $\frac{1}{2 t_n^{(kn)}}$) doesn't necessarily play an enormous role--we could likely choose some 'other way' of determining how much 'budget is left' and push the same choice into Lemma 5.2 (and Theorem 4). Is this correct, or is there something fundamental about the way this function is expressed? If there is nothing fundamental, I would advocate as a reader to abstract over this function, and clarify what properties are 'actually' needed here.

* Nit on conclusion of Thm 4. Can this be stated as a zCDP guarantee? Or are we doing something _specific_ that forces us to go to $(\epsilon, \delta)$-DP?

**Limitations:**

The major unaddressed limitation is the potential difficulty of designing effective algorithms which leverage this perspective and analysis. That is, it is not entirely clear how fruitful the analysis presented here will be. This paper _could_ be _extremely_ strong with some reasonable extensions, perhaps as a theory paper with associated 'mechanism design paper' that provides many instantiations of this theoretical framework to perform various privacy sensitive tasks. For this reason, the paper reads as somewhat borderline to me--potentially pointing the way to serious applications, but not quite closing the gap entirely yet.

Negative social impacts not immediately applicable.

---

> ### Author Rebuttal · Authors · 2023-08-09
>
> Thank you for the thoughtful review.  The goal of the paper is indeed not any particular application to be made precise. Adaptive composition bounds in DP aim to quantify the privacy loss when we make repeated, possibly dependent, private queries (and multiple queries are the norm, nobody would collect a database only so that a single private query can be made of it). The goal in this paper is to generalize the earlier bounds to include ex-post private mechanisms like Brownian noise reduction. This only expands the scope of applicability of the earlier bounds. We are intentionally application-agnostic, and the generality means the results apply to arbitrarily different settings and domains, and they can be adapted to suit the purposes of a variety of companies or situations. The relative error application we write down is a particular natural question of special interest for which the existing solutions are not satisfactory. It is true that one cannot guarantee both a desired level of privacy and a desired level of utility simultaneously, but our methods allow us to tradeoff one against the other and yet bound the privacy loss
>
> Indeed, do note that our final privacy guarantee is in terms of $(\epsilon,\delta)$-DP.  We will clarify this in the abstract. In theorem 3, by sensitivity of a statistic $f_i$ we mean the traditional DP definition of sensitivity, i.e. the maximum change between $f_i(x)$ and $f_i(x’)$ where $x$ and $x’$ are neighboring, no randomness included.  We will clarify this in the next revision.  Note that each noise reduction mechanism uses a single statistic $f_i(x)$ (the true answer) to add noise to and then reduces the noise added to it until some stopping condition. Definition 4.4 gives the Brownian noise reduction that applies a Brownian motion to a single statistic (function on the data).
>
> Sorry for the confusion between the various privacy definitions.  Ex-post privacy mechanisms are the most general, then noise reduction mechanisms are special cases of ex-post private mechanisms.  Then the Brownian noise reduction is a specific instantiation of a noise reduction mechanism (Laplace Noise Reduction is another example of a noise reduction that ensures ex-post privacy). To help clarify, consider independent Gaussian noise with differing standard deviations $\sigma^{(1)}, \cdots, \sigma^{(k)}$ that is added to statistic $f(x)$, then the privacy loss over all noisy terms $L^{(1:k)}$ would not be the same as the privacy loss $L^{(k)}$ for the last noisy term.  However, if we add noise with standard deviation $\sigma^{(k)}$ to $f(x)$ to get $\hat{f}$ we would get privacy loss $L^{(k)}$ and if we then add independent noise terms to $\hat{f}$ with different $\sigma^{(1)}, \cdots, \sigma^{(k-1)}$, then the privacy loss over all terms $L^{(1:k)}$ would be the same as $L^{(k)}$ (because the rest can be viewed as post-processings of $\hat{f}$).
>
> The definition of $(\mathcal{E},\delta, \epsilon^*)$ ex-post-privacy is not the same as $(\epsilon^*,\delta)$-DP.  It is important in our definition that we have both a functional bound on the privacy loss for all of its outcomes and an overall bound on how large the functional bound can get.  Ensuring the $\epsilon^*$ bound prevents using very small noise when not enough privacy budget exists to actually return such a precise value.  This bound is required in Lemma 5.2 and Theorem 4, otherwise a returned result would surpass the remaining budget.  Theorem 4 can be formulated as an approximate zCDP privacy guarantee.  One easy way to do this is that $(\epsilon,\delta)$-DP ensures $\delta$-approximate $\epsilon^2/2$-zCDP [Bun Steinke’16].

---

> > ### Comment · Reviewer_dJHg · 2023-08-13
> >
> > I stand by my overall recommendation of weak accept. The results are, I believe, novel and powerful--but for a mathematically heavy and subtle paper like this one, there are still some presentation and clerity issues that I would prefer to see resolved.
> >
> > I _think_ I would still appreciate some more clarity in the statement of theorem 3 (and indeed in a few other places). E.g., tracing back to understand what is the precise meaning of statistic, I wonder if I see either a small abuse of notation or a typo in definition 4.4. That is: $f$ is declared to be a function with domain $\mathcal{X}$, $T$ a function with domain $\mathcal{Y}^*$, but both are evaluated at the same variable $x$ in the definition of the BM. Definition 4.3 makes me think that there is possibly a suppressed $A$ here, but the discussion of who is a noise reduction mechanism, who is an ex-post-privacy mechanism, etc., makes me think that the BM is implicitly occurring in its own definition? It is possible that sense can be made here, but it is not obvious to me how.
> >
> > A super minor nit: On equation (2), in definition 4.1: I see that this is indeed not identical to epsilon-delta DP. But I think this definition remains _somewhat_ confusing--if it is novel to this paper, I would recommend reconsidering. I believe my confusion here was due to the fact that in some sense $\epsilon^*$ and $\mathcal{E}$ 'specify the same thing'--rather, that $\mathcal{E}$ has strictly _more_ information than $\epsilon^*$. IE, including $\epsilon^*$ in the definition is really just a 'highlighting' of a property of a function $\mathcal{E}$--that it is uniformly bounded above. I believe my mind went to considering equivalence with $(\epsilon, \delta)$ because I found it natural to consider $\mathcal{E}$ as a _variable_--that is, I implicitly read the definition incorrectly, assuming that $\mathcal{E}$ was not _fixed_, but rather $A$ was $(\mathcal{E}, \delta, \epsilon^*)$-ex-post-private if the claimed equation held for all $\mathcal{E}$ which are uniformly bounded above--in this case I think it would indeed be equivalent to $\epsilon, \delta$. So TLDR: I would recommend eliminating the specialization of the notion of ex-post privacy when the function $\mathcal{E}$ has a uniform upper bound, and instead prefer to track the properties of a given $\mathcal{E}$ through arguments (since as far as I can tell this tracking-through-arguments will need to happen in any case).

---

> > > ### Author Response · Authors · 2023-08-16
> > >
> > > Thank you for the helpful comments.  There is a slight abuse of notation with how we are defining the Brownian mechanism.  To help clarify, we can define $BM^{(n)}(x) = (t^{(k)}, f(x) + B^{(t^{(k)})})_{k \leq n}$, and then $BM(x) = BM^{(T(x))}(x)$ where $T(x) = T(BM^{(\infty)}(x))$.  Note that $T(BM^{(\infty)}(x))$ is a stopping time, so that the event $T(BM^{(\infty)}(x))= k$ only needs to know outcomes $BM^{(k-1)}(x)$.  We will clarify this in the next revision.
> > >
> > > Good point about the ex-post privacy definition with three arguments.  The $\epsilon^*$ is indeed a bound on the function $\mathcal{E}()$, so we can keep the original definition of ex-post privacy and set the condition on $\mathcal{E}()$ in the theorem/lemma statements.

---

### Official Review · Reviewer_njjx · 2023-07-05

**Soundness:** 3 good
**Presentation:** 2 fair
**Contribution:** 3 good
**Rating:** 5
**Confidence:** 3

**Summary:**

This paper tries to tackle the following basic question in differential privacy: suppose I desire to answer as many as possible counting queries with low **relative error** with a predetermined privacy budget (\eps,\delta). What is the best mechanism I can use?

Building on prior works on "ex-post" private mechanisms (Whitehouse et al '22), this paper makes the following contributions:
(1) An improved analysis for ex-post privacy mechanisms (in particular the Brownian Noise Reduction Mechanism);
(2) A "Privacy filter" which allows the user to compose concentrated-DP mechanisms and the Brownian mechanism adaptively.
(3) An experimental section, demonstrating the improvement of the proposed algorithm, compared with the baseline approach of "doubling". It appears that, compared with the baseline approach, the new algorithm can answer twice as many queries with the same privacy budget and accuracy requirement.

**Strengths:**

* The new analysis for the Brownian mechanism looks nice. It can potentially make the technique more useful.
* The experiment setup is thoroughly described, and the improvement looks significant.

**Weaknesses:**

* It looks like the composition for ex-post mechanisms is omitted in the main body. However, you claimed this as one of your main contributions.
* Why only analyze the composition of concentrated-DP mechanisms with Brownian mechanisms? This seems to make the theory less general.
* From the practice side, this paper only runs experiments on synthetic data sets, which might not be convincing enough to a broader community.


**Questions:**

* You conjectured the composition for ex-post mechanisms is linear in general. I was wondering why it is the case.
* How does this ex-post private approach compare with the (private) hyper-parameter selection approach? See, for example, here: https://arxiv.org/abs/2301.00301

---

> ### Author Rebuttal · Authors · 2023-08-09
>
> As stated as a general takeaway from the reviewer feedback, we will include the results on the basic, yet more general, composition for ex-post private mechanisms in the next revision.  We presented the composition of (approximate) concentrated DP mechanisms with Brownian mechanisms as we feel that is the most general version of our analysis.  In particular, any $(\epsilon, \delta)$-DP mechanism can be converted to an approximate zCDP mechanism and our advanced composition results (Theorem 4) crucially rely on the only ex-post privacy mechanisms being Brownian noise reductions.  We have not been able to extend this analysis for more general ex-post privacy mechanisms, so we consider that an open problem for future work, although in the meantime we do have the basic composition result.
>
> Although our empirical results are only evaluated on synthetic data, we generate data from a power law distribution, frequently encountered on real datasets.  The empirical results are used to show that the theoretical results can still outperform existing approaches for various sample sizes.  Any real dataset of counts is likely to exhibit a power law distribution and the results will largely be the same on those datasets.  We verified this on Reddit data (see general reviewer comment), where we state the results that we will include in the next revision.
>
> Since we submitted, we realized that the linear general basic composition is the best one could hope for because the ex-post privacy function $\\mathcal{E}_i$ at round $i$ can simply be the actual privacy loss function $L_i$ with independently selected mechanisms $A_i$ at each round, which would lead to the overall ex-post privacy function being the sum of privacy losses.
>
> Thank you for pointing out the work in https://arxiv.org/abs/2301.00301  as this seems to have been a separate line of work from the ex-post privacy literature, although there does seem to be some connection.  The main difference is that they define privacy loss based on the input dataset, rather than the outcome of the algorithm that is done is ex-post privacy.  Furthermore, they consider “local sensitivity” which can be drastically different between different datasets, but in our motivating example of “relative accuracy” the local sensitivity is the same for all datasets (removing one user’s data can reduce counts by at most 1 regardless of the dataset).  Lastly, their generalized PTR approach is attempting to find the “goldilocks” noise scale so that the least amount of noise is added without returning a $\bot$ outcome.  However, the resulting scheme will still satisfy an overall DP guarantee, with predetermined privacy parameters.  There is not a clear way how this could be modified to be composed so that selecting a large noise parameter at some rounds could result in returning more results under an overall DP bound.

---

> > ### Comment · Reviewer_njjx · 2023-08-18
> > **Thank you**
> >
> > Thanks to the authors for answering my questions and concerns. I would like to raise my score by 1, as I think the paper is technically interesting, but the presentation might need some extra effort to polish.

---

### Official Review · Reviewer_hK6N · 2023-07-07

**Soundness:** 3 good
**Presentation:** 1 poor
**Contribution:** 4 excellent
**Rating:** 5
**Confidence:** 2

**Summary:**

‘Accurary-first’ mechanisms try to achieve the strongest possible privacy guarantee that satisfies a target level of accuracy. One such example is a noise reduction mechanism, which scales up the privacy cost / decreases the noise level until the target utility is achieved. Noise reduction mechanisms only pay the privacy cost of the final noise addition, but provide ex-post privacy guarantees whose composition bounds are unknown and which are not compatible with DP. This paper provides composition bounds for ex-post DP, and develops a unified privacy filter that can switch between DP and ex-post DP mechanisms subject to an overall privacy guarantee.

**Strengths:**

1. Very interesting and strong potential for impact. Seems like it could be a big step towards making DP more practical!
2. Novel privacy filter and analysis of the ex-post privacy guarantee for a Brownian filter.

**Weaknesses:**

The paper was very dense and difficult to read, which I feel limits its appeal. (Brownian motion / the Brownian mechanism, for example, are never really explained.) I think the paper would benefit greatly from being gentler to its readers.

**Questions:**

The paper focuses very much on composition bounds for ex-post privacy, but I think that there should also be a discussion on post-processing. The way that Definition 4.1 isn’t defined on sets, it seems like it wouldn’t be closed under post-processing. So I think that converting this into $(\epsilon, \delta)$-DP is an important technical detail that should be discussed in the paper.

On a similar note — why the overloading of notation in Definition 4.1? It was unclear to me which version of ‘ex-post’ was being used throughout the remainder of the paper. And I wasn’t sure of what was the purpose of imposing a bound $\epsilon^*$ on the ex-post privacy loss. Isn’t that just redefining DP but without sets?

**Limitations:**

No issues.

---

> ### Author Rebuttal · Authors · 2023-08-09
>
> While formally introduced in Definition 4.4., we will certainly add a gentler introduction to ex-post or noise-reduction mechanisms like the Brownian mechanism to the paper. For now, an informal description is that it is a recently devised generalization of the Gaussian mechanism: one first adds a “large” Brownian motion noise (lowest utility), and decreases the noise progressively (increasing utility) until a utility-privacy tradeoff is reached; the neat fact is that the total privacy loss of the Brownian mechanism is identical to the Gaussian mechanism at the very last step of the interaction (thus being as if no interaction took place, and paying no price for the interaction).
>
> Regarding post-processing, the definition of ex-post privacy depends on the outcome of the algorithm, not outcomes falling into some set.  So if an algorithm is $(\mathcal{E}(),\delta)$ ex-post private, that does not mean that any post-processing function of the algorithm is also $(\mathcal{E}(),\delta)$, as the post-processing will likely change the outcome distribution.  In general, there is not a generic conversion of ex-post privacy to traditional DP. This is why past papers have either dealt with one notion or the other. Our paper is the first to attempt to bridge this divide and provides an overall DP guarantee even when traditional DP algorithms are used along with ex-post private algorithms.  Our results crucially rely on seeing outcomes of ex-post private mechanisms, not post-processed results, as we need to ensure the privacy loss has the correct distribution.  The analyst may then use the outcomes to determine what to do next (including post-processing it), but the analyst must receive the outcome of the Brownian noise reduction.  The overall DP guarantee of the entire transcript does not get worse if the results of the ex-post private outcomes are post-processed.  The point here is that we are not trying to ensure DP of each outcome, but rather the full interaction.
>
> Introducing the third parameter $\epsilon^*$ to the ex-post private definition does not redefine DP without sets.  It is meant to ensure that the bound on the privacy loss cannot go beyond a certain amount, but the privacy loss still depends on the realized outcome of the algorithm.  This bound on the privacy loss function is important so that no outcome of the algorithm can generate a privacy loss that is very large.  Without the third parameter, $\epsilon^*$ can be interpreted as infinity, so that the privacy loss function $\mathcal{E}$ need not be bounded apriori.

---

> > ### Comment · Reviewer_hK6N · 2023-08-18
> >
> > Thanks for the clarifications. I've raised my score by 1; I don't want to be that one curmudgeonly reviewer holding back this paper from publication, especially as I think the results could be very useful. But like Reviewer njjx, I think that improving the presentation will be a significant under-taking.

---

> > > ### Author Response · Authors · 2023-08-19
> > >
> > > Thank you. We have been revising our paper as we have been responding, and it is indeed already much clearer thanks to the great questions and feedback. We will continue to iterate, especially using the extra page judiciously, if the paper is accepted

---

### Official Review · Reviewer_HnZw · 2023-07-07

**Soundness:** 3 good
**Presentation:** 4 excellent
**Contribution:** 3 good
**Rating:** 7
**Confidence:** 2

**Summary:**

This paper focuses on the study of adaptive compositions of ex-post differential privacy, where privacy guarantees depend on the outcome of the algorithm. The utilization of ex-post DP enables analysts to achieve a more favorable privacy-utility trade-off. The paper introduces a tool for composing ex-post DP and traditional DP, providing a unified privacy filter that combines the Brownian Mechanism with traditional DP mechanisms. The authors apply the proposed framework to the task of releasing counts with bounded relative error and demonstrate empirical improvements over existing methods.

**Strengths:**

The propose framework could be useful for DP practitioners to compose traditional DP and ex-post DP mechanisms. The proposed methods provide DP guarantees subject to outcomes satisfying strict accuracy requirements.


**Weaknesses:**

The presentation of this paper is not very clear to me. The first listed contribution seems to be the general composition theorem for ex-post differential privacy; however, it is only mentioned in the appendix. In the main text, I noticed a list of theorems, but there seems to be a lack of intuition provided, such as explanations on why these bounds are tight.

**Questions:**

In Figure 2 left, why is the standard deviation so large? Is this expected?


**Limitations:**

This paper is theoretical and does not have potential negative societal impact.

---

> ### Author Rebuttal · Authors · 2023-08-09
>
> We do not have a formal lower bound argument for why these composition bounds are tight, but they at least match the “standard” (non-outcome-dependent) privacy loss composition that depends on the variance of the noise that is added.  In traditional DP, if we add noise with standard deviation $1/\epsilon_i$ to each count, we would expect the DP privacy loss to scale with $\sqrt{\sum_i \epsilon_i^2}$, which is similar to the privacy loss composition bounds we develop that allow for noise that can depend on the outcomes. We have now added some more explanation to the paper of why we think the bounds are likely to be tight, not by proving a lower bound, but by analogy to the simpler DP advanced composition.
>
> Figure 2 seems to have a higher variance when the data is small.  This is likely due to how one typically defines “precision” — if a run of the algorithm returns no results, the precision is 100% (the error rate is zero).  Furthermore, when the sample size is small, the choice of the privacy parameter used in the Exponential Mechanism will have a larger impact on the results, as the Exponential Mechanism will likely select from many different counts with similar low counts from small datasets.  Then the Brownian noise reduction may occasionally conclude the small counts have decent relative accuracy when in fact they do not.  We think that developing an “ex-post” privacy version of the Exponential Mechanism is an important direction for future work.

---

> > ### Comment · Reviewer_HnZw · 2023-08-15
> >
> > Thank you for your response and addressing my questions. It would be great if you could add such a discussion near the theorem statements.

---

### Official Review · Reviewer_xyZp · 2023-07-27

**Soundness:** 4 excellent
**Presentation:** 3 good
**Contribution:** 2 fair
**Rating:** 6
**Confidence:** 5

**Summary:**

One of the challenges any attempt to implement DP preserving techniques, is the inherent tradeoff between the level of guaranteed privacy and accuracy levels. While most of the analysis done by privacy experts focused on the perspective of maximizing accuracy subject to an upper bound on the privacy loss, many data scientists might prefer the inverse perspective, e.i., minimizing privacy loss subject to a lower bound on accuracy.
This latter perspective was first formalized by [1], who referred to it as "accuracy first". They proposed a mechanism that releases a sequence of responses with increasing accuracy and decreasing privacy levels, which - somewhat surprisingly - pays only for the privacy loss of the last released response, which they analyze using "ex-post privacy". Later [2] extended these results to Gaussian noise as well using the Brownian mechanism, but since they did not rely on the classical DP notion, they lacked a composition guarantee.

The authors of the current work resolved this issue, by leveraging the privacy filter toolkit, in which privacy parameters can be adaptively chosen, under the constraint that they will not exceed some pre-defined threshold. Specifically, they use a zCDP filter for the Brownian mechanism which can be used for composing several calls to the mechanism as well as other zCDP mechanisms. They use this technique to analyze a setting where the accuracy goal is defined in terms of relative error rather than additive error, and use empirical evaluation to asses the accuracy improvement achieved by applying their analysis technique.

[1] K. Ligett, S. Neel, A. Roth, B. Waggoner, and S. Z. Wu. Accuracy first: Selecting a differential privacy level for accuracy constrained erm.

[2] J. Whitehouse, A. Ramdas, S. Wu, and R. Rogers. Brownian noise reduction: Maximizing privacy subject to accuracy constraints.

**Strengths:**

Providing composition guarantees for ex-post privacy is an important step towered utilizing the accuracy first perspective in the real world, which can help to further extend the usage of differential privacy techniques in the real world. The background and intuition are clearly presented, the results in the body of the paper are formally stated, and the proofs are sound.

**Weaknesses:**

While the results presented in this paper are not just a simple application of the privacy filter technique to the ex-post privacy framework, as discussed in section 5.1, the novel results are somewhat limited in their scope, as they apply only to the zCDP privacy measure and the Brownian mechanism. The empirical evaluation is limited to a specific use case, and a single $(\epsilon, \delta)$ tuple (with $\epsilon = 10$ which is higher than what is usually considered private).

In the introduction, the authors also claim to provide a basic composition theorem for arbitrary ex-post privacy mechanisms, which appears only in the supplementary material. Putting aside the fact that the result does not appear in the body of the paper, and the fact the supplementary material is not comprised of appendix but is an extended version of the paper which makes it hard to find the additions, the actual claim (Theorem 3) is somewhat hard to parse, and could probably benefit from removing the zCDP part.

Small minor comment: It seems like the two references to Whitehouse et al. papers in the abstract were mixed up. The first one should've been the 22 paper and the second the 23.

=======

**Edit after rebuttal discussion:**

The authors have addressed my concerns.

**Questions:**

I have no additional questions to the authors.

---

> ### Author Rebuttal · Authors · 2023-08-09
>
> We removed the basic, yet more general, composition result from the main body of the paper due to space limitations, but it states that the realized privacy parameters add up.  We wanted to highlight the more “advanced” composition bounds that allow for tighter privacy loss bounds but will include the basic composition result in the next revision.  Thank you for pointing out the mixed-up citations in the abstract; we have now corrected this.  We uploaded the full paper+appendices as a single supplement to make it easier to review one file, but in the final version, we will only have the appendices as a supplement. The application setting is meant to show that our theoretical results also improve over existing approaches significantly in practice.  As we mentioned in the general comment to reviewers, we have since ran experiments on real Reddit data and can show similar improvements over the doubling Gaussian approach that we saw on synthetic data even for $(\epsilon = 1, \delta = 10^{-6})$-DP.

---

> > ### Comment · Reviewer_xyZp · 2023-08-14
> >
> > I thank the authors for the clarifications, and have no follow-up questions.

---

### Author Rebuttal · Authors · 2023-08-09

We thank all the reviewers for their thoughtful comments and helpful suggestions.  Many reviewers pointed out that the main paper should have included the basic, yet more general, composition that included zCDP and ex-post privacy mechanisms, which was instead pushed to the Supplement (where it was hard to find). We agree that the basic composition result is interesting in itself and we would have liked to include it in the main body.  We had decided to omit this result from the main body due to space limitations to focus on the “advanced” privacy loss composition as we think that is the most applicable result.  However, given the feedback, we can state the more general result in the main paper (especially if we have an extra page if the paper is accepted).

We have since applied our results to real data, specifically Reddit comments from https://github.com/heyyjudes/differentially-private-set-union/tree/ea7b39285dace35cc9e9029692802759f3e1c8e8/data.  We construct a histogram over all words in the data (including punctuation), eliminating duplicate words from each author.  Hence, each author can contribute an arbitrary number of words, but each word can be contributed at most once.  To help simplify our experiments, we use the top-1000 words in the histogram as the domain of words.  Ideally this domain discovery step would also be done with DP, but that is not the main point of these experiments.  We then follow the same approach in our synthetic data experiments, using the Exponential Mechanism to find the top word from the set of 1000 words and either using the Brownian noise reduction or doubling Gaussian noise approach to reduce the noise until a target 1% relative error.  With a conservative $(\epsilon = 1, \delta = 10^{-6})$-DP guarantee, in 1000 independent trials, more than 95% of the results returned by both methods have the target relative error and the Brownian noise reduction returns about 40% more results than the doubling Gaussian mechanism (152 vs 109 results).  We will include these real data results in the next revision.

---

### Decision · Program_Chairs · 2023-09-21

**Decision:**

Accept (poster)

**Comment:**

This paper addresses adaptive compositions of ex-post differential privacy, where privacy guarantees depend on the outcome of the algorithm (as opposed to being fixed in advance). These mechanisms allow the analyst to prioritize achieving a certain accuracy goal at the lowest privacy loss possible. This work adapts the notion of Renyi filter to allow composing Brownian Mechanism which is ex-post DP and traditional DP mechanims. The authors apply the proposed framework to the task of releasing counts with bounded relative error and demonstrate empirical improvements over existing methods.